# GUIDING CONTINUOUS OPERATOR LEARNING THROUGH PHYSICS-BASED BOUNDARY CONSTRAINTS

**Nadim Saad** [1,†,*] **Gaurav Gupta** [2,†], **Shima Alizadeh** [2], **Danielle C. Maddix** [2]
[1] Stanford University (450 Serra Mall, Stanford, CA 94305)
[2] AWS AI Labs (2795 Augustine Dr, Santa Clara, CA 95054)
[†] Equal contributions, order decided by coin toss.
  nsaad31@stanford.edu, {gauravaz, alizshim, dmmaddix}@amazon.com

## ABSTRACT

Boundary conditions (BCs) are important groups of physics-enforced constraints that are necessary for solutions of Partial Differential Equations (PDEs) to satisfy at specific spatial locations. These constraints carry important physical meaning, and guarantee the existence and the uniqueness of the PDE solution. Current neural-network based approaches that aim to solve PDEs rely only on training data to help the model learn BCs implicitly. There is no guarantee of BC satisfaction by these models during evaluation. In this work, we propose Boundary enforcing Operator Network (BOON) that enables the BC satisfaction of neural operators by making structural changes to the operator kernel. We provide our refinement procedure, and demonstrate the satisfaction of physics-based BCs, e.g. Dirichlet, Neumann, and periodic by the solutions obtained by BOON. Numerical experiments based on multiple PDEs with a wide variety of applications indicate that the proposed approach ensures satisfaction of BCs, and leads to more accurate solutions over the entire domain. The proposed correction method exhibits a (2X-20X) improvement over a given operator model in relative $L^2$ error (0.000084 relative $L^2$ error for Burgers' equation). Code available at: https://github.com/amazon-science/boon.

## 1 INTRODUCTION

Partial differential equations (PDEs) are ubiquitous in many scientific and engineering applications. Often, these PDEs involve boundary value constraints, known as Boundary Conditions (BCs), in which certain values are imposed at the boundary of the domain where the solution is supposed to be obtained. Consider the heat equation that models heat transfer in a one dimensional domain as shown schematically in Figure 1. The left and right boundaries are attached to an insulator (zero heat flux) and a heater (with known heat flux), respectively, which impose certain values for the derivatives of the temperature at the boundary points. No-slip boundary condition for wall-bounded viscous flows, and periodic boundary condition for modeling isotropic homogeneous turbulent flows are other examples of boundary constraints widely used in computational fluid dynamics. Violating these boundary constraints can lead to unstable models and non-physical solutions. Thus, it is critical for a PDE solver to satisfy these constraints in order to capture the underlying physics accurately, and provide reliable models for rigorous research and engineering design.

In the context of solving PDEs, there has been an increasing effort in leveraging machine learning methods and specifically deep neural networks to overcome the challenges in conventional numerical methods (Adler & Öktem, 2017; Afshar et al., 2019; Guo et al., 2016; Khoo et al., 2020; Zhu & Zabaras, 2018). One main stream of neural-network approaches has focused on training models that predict the solution function directly. These methods typically are tied to a specific resolution and PDE parameters, and may not generalize well to different settings. Many of these approaches may learn physical constraints implicitly through training data, and thus do not guarantee their satisfaction at test time (Greenfeld et al., 2019; Raissi et al., 2019; Wang et al., 2021). Some previous works also attempt to formulate these physics-based constraints in the form of a hard constraint

---

*Work completed during internship at AWS AI Labs.

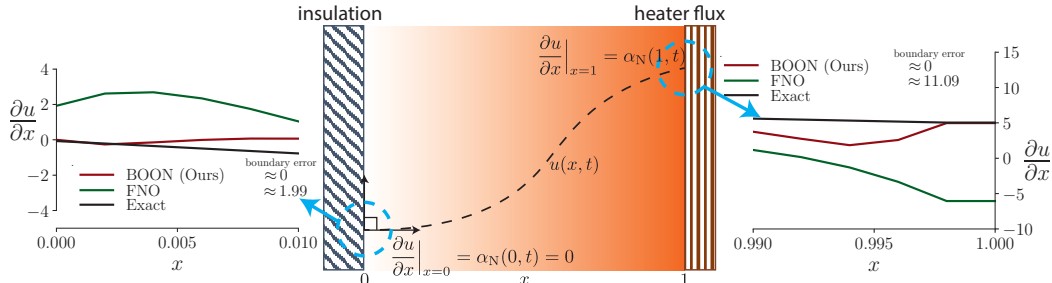

Figure 1: **Heat equation with physical boundary constraints.** Heat flow across a conductor with an insulator and a heater at the left and right boundary, respectively. The insulator physically enforces zero heat flux ($\propto \frac{\partial u}{\partial x}$), and the heater imposes a known heat flux $\alpha_{\mathrm{N}}(1,t)$ from the right boundary. Neumann denotes this derivative-based boundary constraint (see Table 1). Violation of the left boundary constraint by an existing neural operator (FNO), even when trained on data satisfying boundary constraint, suggests heat flowing across the insulator, which is not aligned with the underlying physics. Disagreement at the right boundary violates energy conservation. The proposed boundary corrected model (BOON) produces physically relevant solution along with better overall accuracy (see Section 4.2.1).

optimization problem, although they could be computationally expensive, and may not guarantee model convergence to more accurate results (Xu & Darve, 2020; Krishnapriyan et al., 2021; Lu et al., 2021b; Donti et al., 2021).

Neural Operators (NOs) are another stream of research which we pursue in our study here that aim to learn the operator map without having knowledge of the underlying PDEs (Li et al., 2020a; Rackauckas et al., 2020; Tran et al., 2021; Li et al., 2020b; Guibas et al., 2021; Bhattacharya et al., 2021; Gupta et al., 2021). NO-based models can be invariant to PDE discretization resolution, and are able to transfer solutions between different resolutions. Despite the advantages NO-based models offer, they do not yet guarantee the satisfaction of boundary constraints. In Figure 1, the variation of heat flux is shown at the boundary regions for a vanilla NO-based model (FNO) and our proposed model (BOON). FNO violates both boundary constraints, and results in a solution which deviates from the system underlying physics.

Physics-based boundary conditions are inherent elements of the problem formulation, and are readily available. These constraints are mathematically well-defined, and can be explicitly utilized in the structure of the neural operator. We show that careful leverage of this easily available information improves the overall performance of the neural operator. We propose Boundary enforcing Operator Network (BOON), which allows for the use of this physical information. Given an integral kernel-based NO representing a PDE solution, a training dataset $\mathcal{D}$, and a prescribed BC, BOON applies structural corrections to the neural operator to ensure the BC satisfaction by the predicted solution.

Our main contributions in this work can be summarized as follows: **(i)** A systematic change to the kernel architecture to guarantee BC satisfaction. **(ii)** Three numerically efficient algorithms to implement BC correction in linear space complexity and no increase in the cost complexity of the given NO while maintaining its resolution-independent property. **(iii)** Proof of error estimates to show bounded changes in the solution. **(iv)** Experimental results demonstrating that our proposed BOON has state-of-the-art performance on a variety of physically-relevant canonical problems ranging from 1D time-varying Burgers' equation to complex 2D time-varying nonlinear Navier-Stokes lid cavity problem with different BC, e.g. Dirichlet, Neumann, and periodic.

## 2 TECHNICAL BACKGROUND

Section 2.1 formally describes the boundary value problem. Then, we briefly describe the operators in Sections 2.2 and 2.3, which are required for developing our solution in Section 3.

### 2.1 BOUNDARY VALUE PROBLEMS

Boundary value problems (BVPs) are partial differential equations (PDEs), in which the solutions are required to satisfy a set of constraints along given spatial locations. Primarily, the constraints assign physical meaning, and are required for uniqueness of the solution. Formally, a BVP is written as:

$$\left.\begin{array}{l} \mathcal{F}u(x,t) = 0, \ x \in \Omega \subset \mathbb{R}^n, \\ u(x,0) = u_0(x), \ x \in \Omega, \\ \mathcal{G}u(x,t) = 0, \ x \in \partial\Omega, \end{array}\right\}, \forall t \geq 0, \tag{1}$$

where $\mathcal{F}$ denotes a time-varying differential operator (non-linear, in general), $\Omega$ the spatial domain with boundary $\partial\Omega$, $u_0(x)$ the initial condition, $\mathcal{G}$ the boundary constraint operator, and $u(x,t)$ the solution at time $t$. Formally, given the domain $\Omega$, we denote the interior as $\overset{\circ}{\Omega}$, which is the space of points $x \in \Omega$ such that there exists an open ball centered at $x$ and completely contained in $\Omega$. The boundary of $\Omega$ is defined as $\partial\Omega = \{x | x \in \Omega \ \cap \ x \notin \overset{\circ}{\Omega}\}$. In a given boundary domain $\partial\Omega$, the boundary conditions (BCs) are specified through $\mathcal{G}u = 0$, where $\mathcal{G}$ is taken to be linear in the current work. Three types of BCs which are widely used to model physical phenomena are: (1) Dirichlet, (2) Neumann, and (3) periodic. Table 1 describes the mathematical definition of each boundary condition.

| Boundary condition | Description | $\partial\mathcal{U}$ |
|---|---|---|
| Dirichlet | Fixed-value at the boundary | $\{u(x,t) | \mathcal{G}_{\mathrm{D}}u(x_d,t) := u(x_d,t) - \alpha_{\mathrm{D}}(x_d,t) = 0, x_d \in \partial\Omega, \alpha_{\mathrm{D}}(x_d,t) \in \mathbb{R}, \forall x \in \Omega, \forall t \geq 0\}$ |
| Neumann | Fixed-derivative at the boundary | $\{u(x,t) | \mathcal{G}_{\mathrm{N}}u(x_d,t) := \frac{\partial u}{\partial x}(x_d,t) - \alpha_{\mathrm{N}}(x_d,t) = 0, x_d \in \partial\Omega, \alpha_{\mathrm{N}}(x_d,t) \in \mathbb{R}, \forall x \in \Omega, \forall t \geq 0\}$ |
| Periodic | Equal values at the boundary | $\{u(x,t) | \mathcal{G}_{\mathrm{P}}u(x_0,t) := u(x_0,t) - u(x_{N-1},t) = 0, x_0, x_{N-1} \in \partial\Omega, \forall x \in \Omega, \forall t \geq 0\}$ |

Table 1: **Boundary conditions.** Examples of the most useful canonical boundary conditions, i.e. Dirichlet, Neumann, and periodic, which carry different physical meanings. We define $\partial\mathcal{U}$ to be the function space corresponding to a given boundary condition. Illustration on a one-dimensional spatial domain $\Omega = [x_0, x_{N-1}]$ with boundary $\partial\Omega = \{x_0, x_{N-1}\}$ is shown for simplicity, and an extension to a high-dimensional domain is straight-forward (see Section 4).

## 2.2 OPERATORS

The solution of a PDE can be represented as an operator map, from either a given initial condition, forcing function, or even PDE coefficients (input $a$) to the final solution $u$. Formally, an operator map $T$ is such that $T : \mathcal{A} \to \mathcal{U}$, where $\mathcal{A}$ and $\mathcal{U}$ denote two Sobolev spaces $\mathcal{H}^{s,p}$, with $s > 0$. Here, we choose $p = 2$. For simplicity, when $T$ is linear, the solution can be expressed as an integral operator with kernel $K : \Omega \times \Omega \to L^2$:

$$u(x,t) = Ta(x) = \int_\Omega K(x,y)a(y)dy, \tag{2}$$

where $a(x) \in \mathcal{A}$, $u(x,t) \in \mathcal{U}$ with specified $t > 0$. In this work, $a(x)$ is equivalent to the initial condition, $u_0(x)$ in eq. (1).

**Note:** For a given PDE, the operator $T$ is non-linear in general, and we concatenate multiple linear operators along with non-linearities (e.g., GeLU) to model a non-linear operator (as done in Li et al. (2020a), Gupta et al. (2021)). We refer to Section 4 for more details.

## 2.3 BOUNDARY-CONSTRAINED OPERATORS

Consider a BVP of the form in eq. (1), we denote the true operator between initial condition to the solution at time $t$ as $\mathcal{P} : \mathcal{A} \to \mathcal{U}_{\mathrm{bdy}}$ with $\mathcal{U}_{\mathrm{bdy}} = \mathcal{U} \cap \partial\mathcal{U}$, where $\mathcal{U}_{\mathrm{bdy}}$ denotes the function space of solutions to the BVP, as illustrated in Figure 2 (See Appendix A for a summary of the notations). For an operator $T : \mathcal{A} \to \mathcal{U}_{\mathrm{bdy}}$ (learned from the data), the incurred boundary error is denoted as

$$\mathcal{E}_{\mathrm{bdy}}(T) = \frac{1}{|\mathcal{D}|} \sum_{(a,.)\in\mathcal{D}} ||\mathcal{G}T(a) - \mathcal{G}\mathcal{P}(a)||_{\partial\mathcal{H}^{s,p}}, \tag{3}$$

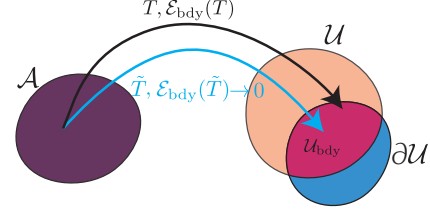

Figure 2: **Operator refinement.** A given operator $T$ and its refinement $\tilde{T}$.

where $||.||_{\partial\mathcal{H}^{s,p}}$ is taken as the $L^2$ error at the boundary, or $(\int_{\partial\Omega} ||.||^2 dx)^{1/2}$, $\mathcal{D} \subset (\mathcal{A}, \mathcal{U}_{\mathrm{bdy}})$ is a given dataset (for example, evaluation data), and $\mathcal{G}$ is a linear boundary operator (see eq. (1)). Although the operator $T$ is trained using the data with samples drawn from $\mathcal{U}_{\mathrm{bdy}}$, it is not guaranteed to have small boundary errors over the evaluation dataset. In this work, we aim to develop a correction

mechanism such that the resulting operator $\tilde{T} : \mathcal{A} \to \mathcal{U}_{\text{bdy}}$ satisfies $\mathcal{E}_{\text{bdy}}(\tilde{T}) < \epsilon$ over given data $\mathcal{D}$, for an arbitrary small $\epsilon > 0$.

**Remark**: Emphasizing small boundary error through model modification serves two purposes, first (i) satisfaction of boundary values which are necessary for physical importance of the problem, second and more importantly (ii) to act as guiding mechanism for obtaining a correct solution profile (see Section 4 and Figure 3).

## 3    BOON: Making kernel corrections for operator

We have seen in our motivating example in Figure 1 that even though a neural operator is provided with the training data which implicitly obeys the boundary constraints, the boundary conditions may be violated in the final solution. Explicit knowledge of the given boundary conditions can be exploited to guide the solution profiles, and ensure lower boundary errors in the evaluation. Section 3.1 provides motivation for an explicit BC satisfaction procedure, and Section 3.2 discusses the computational algorithms for an efficient implementation.

### 3.1    Motivation: Kernel manipulation

The integral kernel $K(x, y)$ in eq. (2) is the fundamental block in operator learning. We observe that the boundary conditions in Table 1 can be exactly satisfied by an appropriate kernel adjustment using the following result.

**Proposition 1.** *For an integral operator $T : \mathcal{H}^{s,p} \to \mathcal{H}^{s,p}$ with kernel $K(x, y)$ mapping an initial condition $u_0(x)$ to $u(x, t)$, the boundary conditions listed in Table 1 are satisfied, if*

1. *Dirichlet: $K(x_0, y) = \frac{\alpha_D(x_0, t)}{\alpha_D(x_0, 0)} \mathbb{1}(y = x_0)$, $x_0 \in \partial\Omega$, $\forall y \in \Omega \Rightarrow u(x_0, t) = \alpha_D(x_0, t)$.*
2. *Neumann: $K_x(x_0, y) = \frac{\alpha_N(x_0, t)}{u_0(x_0)} \mathbb{1}(y = x_0)$, $x_0 \in \partial\Omega$, $\forall y \in \Omega \Rightarrow u_x(x_0, t) = \alpha_N(x_0, t)$.*
3. *Periodic: $K(x_0, y) = K(x_{N-1}, y)$, $x_0, x_{N-1} \in \partial\Omega$, $\forall y \in \Omega \Rightarrow u(x_0, t) = u(x_{N-1}, t)$.*

The corner cases, $\alpha_D(x_0, 0) = 0$ for Dirichlet and $u_0(x_0) = 0$ for Neumann, are considered in the proof of Proposition 1 in Appendix B. Proposition 1 motivates a key idea that kernel manipulation can propagate certain properties, e.g. boundary constraints from the input function $u_0(x)$ ($a(x)$ in eq. (2)) to the output $u(x, t), t > 0$. For example, if the input $u_0(x)$ is periodic, or $u_0(x_0) = u_0(x_{N-1}), x_0, x_{N-1} \in \partial\Omega$, then changing the kernel through Proposition 1 ensures that $u(x_0, t) = u(x_{N-1}, t), t > 0$ as well, i.e. the output is periodic. Since boundary conditions need to be satisfied by all $u(x, t), \forall t$ (see formulation in eq. (1)), we can leverage kernel manipulation to communicate boundary constraints from the known input function at $t = 0$ to the unknown output at $t > 0$.

### 3.2    Kernel correction algorithms

The previous discussion motivates that we should perform a kernel manipulation for boundary constraint satisfaction. From a computational perspective, the input and the output functions are evaluated over a discretized grid (see Section 4). We show that Gaussian elimination followed by normalization allows for Dirichlet and Neumann, and weighted averaging allows for periodic boundary corrections. We establish the boundary correction of a given neural operator (or BOON) using the following result.

**Theorem 1.** *Refinement: Given a grid with resolution $N$, and an operator $T : \mathcal{A} \to \mathcal{U}_{bdy}$, characterized by the kernel $K(., .)$ as in eq. (2). For every $\partial\mathcal{U}$ in Table 1, there exists a kernel correction such that the resulting operator $\tilde{T} : \mathcal{A} \to \mathcal{U}_{bdy}$ satisfies $\mathcal{E}_{bdy}(\tilde{T}) \approx 0$, evaluated over the grid.*

Figure 5 (in Appendix C) intuitively explains the existence of a solution for Theorem 1, where for a given grid resolution $N$, the discretized version $\mathbf{K} = [K_{i,j}]_{0:N-1,0:N-1} \in \mathbb{R}^{N \times N}$, shown in the blue grid, of the continuous kernel $K(x, y)$ is corrected by applying transformations for the boundary corrections. See Appendix C.1 for the choice and correctness of these transformations.

To efficiently implement our BOON boundary correction, we utilize a given module $\mathcal{K}$, such that $\mathbf{y} = \mathcal{K}(\mathbf{x}) := \mathbf{K}\mathbf{x}$. In Appendix C.2, we show that matrix vector multiplications are the only operations that our kernel correction requires. As common with other Neural Operator implementations, we take $\mathcal{K}$, for example to be Fourier-based in Li et al. (2020a), or multiwavelet-based in Gupta et al. (2021),

to efficiently perform these matrix vector multiplications. The computational efficiency of BOON is established through the following result.

**Theorem 2.** *Computation complexity. Given a grid resolution $N$, a kernel evaluation scheme $\mathcal{K}$ for eq. (2) with cost $N_\mathcal{K}$, the kernel corrections in Theorem 1 can be reduced to at most 3 kernel calls, requiring only $O(N_\mathcal{K})$ operations and $O(N)$ space.*

The proof is provided in Appendix C.2, where we provide arithmetic expansions of the terms in Appendix C.1. These derivations lead to Algorithms 1, 2, 3 for Dirichlet, Neumann, and periodic corrections, respectively. A straight-forward translation of these algorithms to a neural operator architecture with module $\mathcal{K}$ is provided in the Figure 6 (see Appendix C.3).

| **Algorithm 1:** Dirichlet | **Algorithm 2:** Neumann | **Algorithm 3:** Periodic |
|---|---|---|
| **Input:** $\mathcal{K}, \mathbf{u}_0, \alpha_D(x_0, t)$ 
 **Output:** Corrected Dir $\mathbf{u}(t)$ 
 1: $K_{0,0} \leftarrow \mathcal{K}(\mathbf{e}_0)[0]$, with $\mathbf{e}_0 = [1, 0, 0, ...]^T$ 
 2: $\mathbf{z} \leftarrow \mathcal{K}(\mathbf{u}_0)$ 
 3: $\mathbf{u} \leftarrow \mathbf{u}_0$ 
 4: $\mathbf{u}[0] \leftarrow 2\mathbf{u}_0[0] - \mathbf{z}[0]/K_{0,0}$ 
 5: $\mathbf{u} \leftarrow \mathcal{K}(\mathbf{u})$ 
 6: $\mathbf{u}[0] \leftarrow \alpha_D(x_0, t)$ | **Input:** $\mathcal{K}, \mathbf{u}_0, \alpha_N(x_0, t),$ 
 $\quad \mathbf{c} \in \mathbb{R}^N, c_0 \neq 0$ 
 **Output:** Corrected Neu $\mathbf{u}(t)$ 
 1: $K_{0,0} \leftarrow \mathcal{K}(\mathbf{e}_0)[0]$, with $\mathbf{e}_0 = [1, 0, 0, ...]^T$ 
 2: $\mathbf{z} \leftarrow \mathcal{K}(\mathbf{u}_0)$ 
 3: $\mathbf{u} \leftarrow \mathbf{u}_0$ 
 4: $\mathbf{u}[0] \leftarrow 2\mathbf{u}_0[0] - \mathbf{z}[0]/K_{0,0}$ 
 5: $\mathbf{u} \leftarrow \mathcal{K}(\mathbf{u})$ 
 6: $\mathbf{u}[0] \leftarrow \alpha_N(x_0, t)/c_0$ 
 $\quad - \sum_{k>0}(c_k/c_0)\mathbf{u}[k]$ | **Input:** $\mathcal{K}, \mathbf{u}_0, \alpha, \beta \in \mathbb{R}^+,$ 
 $\quad \alpha + \beta = 1$ 
 **Output:** Corrected Per $\mathbf{u}(t)$ 
 1: $\mathbf{u} \leftarrow \mathcal{K}(\mathbf{u}_0)$ 
 2: $\mathbf{u}[0] \leftarrow \alpha\mathbf{u}[0] + \beta\mathbf{u}[-1]$ 
 3: $\mathbf{u}[-1] \leftarrow \mathbf{u}[0]$ |

**Note**: If the given kernel module $\mathcal{K}$ is grid resolution-independent, i.e., train and evaluation at different resolution, then the boundary correction algorithms 1,2, and 3 are resolution-independent as well by construction.

Lastly, we show that the kernel correction in Theorem 1, implemented to guide the operator $T$ through boundary conditions, does not yield unrestricted changes to solution in the interior using the following result which is proved in Appendix C.4.

**Theorem 3.** *Boundedness: For any $a \in \mathcal{A}$, and an operator $T$, with its refinement $\tilde{T}$ (Theorem 1), such that $u = Ta$ and $\tilde{u} = \tilde{T}a$, $||u - \tilde{u}||_2$ is bounded.*

Interestingly, when the given operator $T$ is FNO, then Theorem 3 can be combined with the results of Kovachki et al. (2022) to bound the error between the refined operator $\tilde{T}$ and the true solution.

With a straightforward modification, the refinement results and algorithms presented for a 1D domain in this section can be extended to higher dimensions (and multi-time step predictions) as we demonstrate empirically in Section 4.

## 4 EMPIRICAL EVALUATION

In this section, we evaluate the performance of our BOON model on several physical BVPs with different types of boundary conditions.

For learning the operator, we take the input as $a(x) \coloneqq u_0(x)$ in eq. (1), and the output as $T : u_0(x) \rightarrow [u(x, t_1), \ldots, u(x, t_M)]$ for the given ordered times $t_1 < \ldots < t_M \in \mathbb{R}_+$. For single-step prediction, we take $M = 1$, and for multi-step prediction, we take $M = 25$.

**Data generation:** We generate a dataset $\mathcal{D} = \{(a_i, u_{i,j})_k\}_{k=1}^{n_{\text{data}}}$, where $a_i := a(x_i)$, $u_{i,j} := u(x_i, t_j)$ for $x_i \in \Omega$, $i = 0, \ldots, N - 1$, and $t_j < t_{j+1} \in \mathbb{R}_+$, $j = 1, \ldots, M$ for a given spatial grid-resolution $N$, and number of output time-steps $M$. Each data sample has size $(N, N \times M)$ (See Appendix D). The train and test datasets, $\mathcal{D}_{\text{train}}$ and $\mathcal{D}_{\text{test}}$ are disjoint subsets of $\mathcal{D}$ with sizes $n_{\text{train}}$ and $n_{\text{test}}$, respectively (See Table 7).

**Benchmark models:** We compare BOON model with the following baselines: (i) Fourier neural operator (FNO) (Li et al., 2020a), (ii) physics informed neural operator (PINO) (Li et al., 2021) in

the multi-step prediction case or our adapted PINO (APINO) in the one-step prediction case [1], (iii) multi-pole graph neural operator (MGNO) (Li et al., 2020c), and (iv) DeepONet (Lu et al., 2021a). Note that MGNO requires a symmetric grid, and is valid only for single-step predictions. We use FNO as the kernel-based operator in our work, and MGNO and DeepONet as the non-kernel based operators. See Appendix F.5 for a study of applying BOON to the multiwavelet-based neural operator (Gupta et al., 2021).

**Training:** The models are trained for a total of 500 epochs using Adam optimizer with an initial learning rate of 0.001. The learning rate decays every $50/100$ epochs (1D/rest) with a factor of 0.5. We use the relative-$L^2$ error as the loss function (Li et al. (2020a)). We compute our BOON model by applying kernel corrections (see Section 3) on four stacked Fourier integral operator layers with GeLU activation (Li et al. (2020a)) (See Table 8). We use a p3.8xlarge Amazon Sagemaker instance (Liberty et al., 2020) in the experiments. See Appendix E for details.

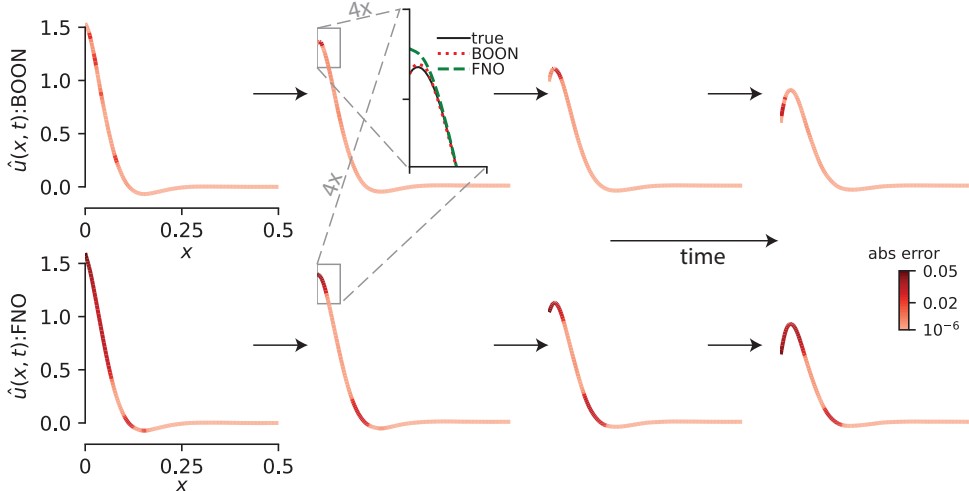

Figure 3: **Multi-time step prediction with boundary correction**: The predicted output $\hat{u}(x, t)$ for BOON (**top**) and FNO (**bottom**). Four uniformly-spaced time snapshots (from a total 25, see Section 4.1.1) are shown from left to right in the spatial domain $\Omega = [0, 0.5]$. The color indicates absolute error at each spatial location. FNO has high boundary error which propagates to the interior, and impacts the overall solution profile. BOON (FNO with boundary correction) uses boundary values to better learn the overall output across the entire time-horizon. Critical boundary mistake is visualized in a 4x zoomed window, where BOON correctly learns the trend.

## 4.1 DIRICHLET BOUNDARY CONDITION

We here present our experimental results for BVPs with Dirichlet boundary conditions including one-dimensional Stokes' second problem (2D including time), one-dimensional Burgers' equation (2D including time), and two-dimensional lid-driven Cavity Flow (3D including time).

### 4.1.1 1D STOKES' SECOND PROBLEM

The one-dimensional Stokes' second problem is given by the simplified Navier-Stokes equation (White (2006)) in the $y-$coordinate as follows,

$$
\begin{aligned}
u_t &= \nu u_{yy}, & y &\in [0, 1], t \geq 0, \\
u_0(y) &= U e^{-ky} \cos(ky), & y &\in [0, 1], \\
u(y = 0, t) &= U \cos(\omega t), & t &\geq 0,
\end{aligned}
\tag{4}
$$

---

[1]PINO (Li et al., 2021) uses a PDE soft constraint from Raissi et al. (2019), which only supports the multi-step prediction case. For single step prediction, we use our adapted APINO, which only uses the boundary loss term.

where $U \geq 0$ denotes the maximal speed of the oscillatory plate located at $y = 0$, $\nu \geq 0$ denotes the viscosity, $\omega$ denotes the oscillation frequency, and $k = \sqrt{\omega/(2\nu)}$ (See Appendix D.1 for the parameter values). We compute the multi-step prediction with final time $t_M = 2$ for $M = 25$. We also apply an exponential mollifier after the last layer in the model's architecture to avoid numerical oscillations in the boundary regions.

| Model | $\nu = 0.1$ | $\nu = 0.02$ | $\nu = 0.005$ | $\nu = 0.002$ | $\nu = 0.001$ |
|---|---|---|---|---|---|
| BOON (Ours) | **0.0089 (0)** | **0.0105 (0)** | **0.0075 (0)** | **0.0164 (0)** | **0.0183 (0)** |
| FNO | 0.0099 (0.0074) | 0.0158 (0.0072) | 0.0188 (0.0084) | 0.0211 (0.0105) | 0.0273 (0.0135) |
| PINO | 0.0241 (0.0841) | 0.0256 (0.0578) | 0.0221 (0.0342) | 0.0188 (0.0223) | 0.0191 (0.0175) |
| DeepONet | 0.1471 (0.0556) | 0.1047 (0.0382) | 0.1408 (0.0515) | 0.1335 (0.0488) | 0.1538 (0.0549) |

Table 2: **Multi-step prediction for Stokes' with Dirichlet BC**. Relative $L^2$ error ($L^2$ boundary error) for Stokes' second problem with varying viscosities $\nu$ at resolution $N = 500$ and $M = 25$.

Table 2 shows that our boundary constrained method results in the smallest relative $L^2$ errors across various values of the viscosity $\nu$ by enforcing the exact boundary values. Figure 3 illustrates that our BOON model corrects the solution profile near the boundary area, and results in a more accurate solution in the interior of the domain. This observation demonstrates that the satisfaction of boundary constraints is necessary to predict an accurate solution that is consistent with the underlying physics of the problem.

### 4.1.2   1D Burgers' Equation

The one-dimensional viscous Burgers' equation with Dirichlet boundary conditions can be formulated as follows:

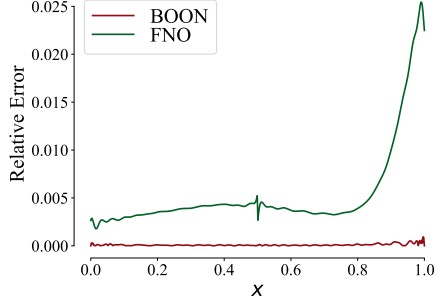

$$u_t + (u^2/2)_x = \nu u_{xx}, \qquad x \in [0,1], t \geq 0,$$
$$u_0(x) = \begin{cases} u_L, & \text{if } x \leq 0.5, \\ u_R, & \text{if } x > 0.5, \end{cases} \quad x \in [0,1], \tag{5}$$
$$u(0,t) = u_{\text{exact}}(0,t), \qquad t > 0,$$
$$u(1,t) = u_{\text{exact}}(1,t), \qquad t > 0,$$

where $\nu \geq 0$ denotes the viscosity, $u_{\text{exact}}(x,t)$ is the exact solution given in eq. (53) and $u_L > u_R \in \mathbb{R}$. For the initial condition values given in Appendix D.2.1, the exact solution involves the formation of sharp interface, known as a shock wave, which propagates towards the right boundary. We compute the single-step prediction at final time $t_M = 1.2$ ($M = 1$) when the shock hits the right boundary. Figure 4 indicates that the relative error of FNO is higher than BOON model throughout the domain and it grows significantly near the right boundary $x_{N-1} = 1$ at $t_M = 1.2$, while the relative error of BOON model stays relatively flat. Table 3 represents the $L^2$ errors obtained by our model as well as the benchmark models. BOON model exhibits a $4X - 30X$ improvement in accuracy, and satisfies the prescribed boundary conditions exactly across various values of the viscosity $\nu$.

Figure 4: Relative error vs $x$ for Burgers' equation with $\nu = 0.02$. The worst test error sample for BOON is shown.

| Model | $\nu = 0.1$ | $\nu = 0.05$ | $\nu = 0.02$ | $\nu = 0.005$ | $\nu = 0.002$ |
|---|---|---|---|---|---|
| BOON (Ours) | **0.00012 (0)** | **0.00010 (0)** | **0.000084 (0)** | **0.00010 (0)** | **0.00127 (0)** |
| FNO | 0.0037 (0.0004) | 0.0034 (0.0004) | 0.0028 (0.0005) | 0.0043 (0.0004) | 0.0050 (0.0022) |
| APINO | 0.0039 (0.0004) | 0.0040 (0.0005) | 0.0037 (0.0007) | 0.0048 (0.0008) | 0.006 (0.0016) |
| MGNO | 0.0045 (0.0004) | 0.0046 (0.0005) | 0.0048 (0.0008) | 0.0064 (0.0015) | 0.0101 (0.0006) |
| DeepONet | 0.00587 (0.00124) | 0.0046 (0.00130) | 0.00580 (0.00128) | 0.00623 (0.00140) | 0.00773 (0.00141) |

Table 3: **Single-step prediction for Burgers' with Dirichlet BC**. Relative $L^2$ test error ($L^2$ boundary error) for Burgers' equation with varying viscosities $\nu$ at resolution $N = 500$ and $M = 1$.

### 4.1.3 2D NAVIER-STOKES EQUATION: LID-DRIVEN CAVITY FLOW

The two-dimensional Navier-Stokes equation for a viscous and incompressible fluid with no-slip boundary conditions can be written in terms of vorticity as follows:

$$
\begin{aligned}
w_t + \vec{u} \cdot \nabla w &= (1/Re)\nabla^2 w, & x, y \in [0,1]^2, t \geq 0, \\
\nabla \cdot \vec{u} &= 0, & x, y \in [0,1]^2, t \geq 0, \\
w_0(x,y) &= \nabla \times \left( \begin{bmatrix} U \\ 0 \end{bmatrix} \mathbb{1}(y=1) \right), & x, y \in [0,1]^2, \quad (6) \\
w(x,0,t) &= -u_y^{(1)}(x,0,t), \; w(x,1,t) = -u_y^{(1)}(x,1,t), & x \in [0,1], t \geq 0, \\
w(0,y,t) &= u_x^{(2)}(0,y,t), \; w(1,y,t) = u_x^{(2)}(1,y,t), & y \in [0,1], t \geq 0,
\end{aligned}
$$

where $\vec{u} = \begin{bmatrix} u^{(1)} & u^{(2)} \end{bmatrix}^T$ denotes the 2D velocity vector, $w = (\nabla \times \vec{u}) \cdot [0,0,1]^T$ denotes the $z$ component of the vorticity, $Re$ the Reynolds number and $U > 0$. See Appendix D.3 for the generation of the reference solution, and the choice of the different parameters. We compute the multi-step prediction for $M = 25$ with final time $t_M = 2$. According to Table 4, BOON results in the lowest errors compared to state-of-the-art models for multiple values of the Reynolds numbers, and additionally reduces the boundary errors to absolute 0.

| Model | $Re = 10$ | $Re = 100$ | $Re = 1000$ |
|---|---|---|---|
| BOON (Ours) | **0.00213 (0)** | **0.00143 (0)** | **0.00176 (0)** |
| FNO | 0.00249 (0.7489) | 0.00186 (0.6022) | 0.00215 (0.9315) |
| PINO | 0.00723 (0.6582) | 0.00532 (0.4294) | 0.00693 (0.7128) |

Table 4: **Multi-step prediction for Navier-Stokes with Dirichlet BC**. Relative $L^2$ error ($L^2$ boundary error) for various benchmarks on the lid-driven cavity problem with varying Reynolds numbers $Re$ at resolution $N = 100$ and $M = 25$.

### 4.2 NEUMANN BOUNDARY CONDITION

In this subsection, we present our experimental results for BVPs with Neumann boundary conditions considering one-dimensional heat equation (2D including time), and two-dimensional wave equation (3D including time). We use finite difference scheme to estimate the given Neumann boundary conditions and enforce them in our operator learning procedure (See Lemma 2).

### 4.2.1 1D HEAT EQUATION

The one-dimensional inhomogenous heat equation is given as:

$$
\begin{aligned}
u_t - k u_{xx} &= f(x,t), & x \in [0,1], t \geq 0, \\
u_0(x) &= \cos(\omega \pi x), & x \in [0,1], \quad (7) \\
u_x(0,t) &= 0, \; u_x(1,t) = U \sin \pi t, & t \geq 0,
\end{aligned}
$$

where $k > 0$ denotes the conductivity, $f(x,t) = U\pi \frac{x^2}{2} \cos \pi t$ denotes the heat source, and $U, \omega \in \mathbb{R}$. The exact solution and more details about the choice of different parameters can be found in Appendix D.4. We compute the multi-step solutions of the heat equation with final time $t_M = 2$ and $M = 25$. Table 5 shows that BOON performs significantly better than the benchmark models in terms of both relative $L^2$ and the $L^2$ boundary errors. In particular, the large boundary errors of baseline models indicates their violation of the physical constraints of the problem, the zero flux on the left boundary ($x = 0$) imposed by the insulator, and the conservation of system total energy that is controlled by the heat source on the right boundary ($x = 1$). As illustrated in Figure 1, by enforcing the exact boundary constraints, BOON corrects the solution profile near the boundary and leads to more accurate result in the interior domain as well.

| Model | $N = 50$ | $N = 100$ | $N = 250$ | $N = 500$ |
|---|---|---|---|---|
| BOON (Ours) | **0.00675 (0)** | **0.00501 (0)** | **0.00893 (0)** | **0.01187 (0)** |
| FNO | 0.00915 (0.96924) | 0.00845 (0.60455) | 0.00925 (2.05607) | 0.01239 (6.31774) |
| PINO | 0.01155 (1.23922) | 0.00829 (0.25646) | 0.06361 (0.01228) | 0.07171 (0.03457) |
| DeepONet | 0.10323 (7.1092) | 0.11995 (10.1063) | 0.18961 (16.6690) | 0.27444 (24.3496) |

Table 5: **Multi-step prediction for the heat equation with Neumann BC**. Relative $L^2$ error ($L^2$ boundary error) for various benchmarks with varying resolutions $N$ and $M = 25$.

### 4.2.2 2D WAVE EQUATION

The wave equation models the vibrations of an elastic membrane with Neumann BC at the boundary points (Cannon & Dostrovsky (1981)). The mathematical formulation of this problem is as follows:

$$
\begin{aligned}
u_{tt} &= c^2(u_{xx} + u_{yy}), & x, y &\in [0,1]^2, t \geq 0, \\
u_0(x,y) &= k\cos(\pi x)\cos(\pi y), & x, y &\in [0,1]^2, \\
u_t(x,y,0) &= 0, & x, y &\in [0,1]^2, \\
u_x(0,y,t) &= u_x(1,y,t) = u_y(x,0,t) = u_y(x,1,t) = 0, & t &\geq 0,
\end{aligned}
\tag{8}
$$

where the parameters $c \in \mathbb{R}_{>0}$ denotes the wavespeed and $k \in \mathbb{R}$. The details of our experiment setup can be found in Appendix D.5. We compute the multi-step solution of the wave equation at final time $t_M = 2$ with $M = 25$ Table 6 shows that BOON obtains the lowest relative $L^2$ error compared to the benchmark models, and enforces the exact physical constraints at the boundary points while capturing the displacement of the membrane with the highest accuracy.

| Model | $N = 25$ | $N = 50$ | $N = 100$ |
|---|---|---|---|
| BOON (Ours) | **0.00097 (0)** | **0.000893 (0)** | **0.00096 (0)** |
| FNO | 0.00140 (0.04296) | 0.00093 (0.03402) | 0.00119 (0.03350) |
| PINO | 0.00113 (1.2873) | 0.00526 (0.24901) | 0.00635 (0.03572) |

Table 6: **Multi-step prediction for the wave equation with Neumann BC**. Relative $L^2$ error ($L^2$ boundary error) for various benchmarks with varying resolutions $N$ and $M = 25$.

### 4.3 ADDITIONAL RESULTS

See Appendix F for the following additional experiments.

**Single-step and multi-step predictions:** We include the corresponding single-step prediction results for Stokes' (Dirichlet), Burgers' (periodic) and heat (Neumann) in Tables 9, 11, 13, respectively. We also include the multi-step time-varying prediction results for the Riemann problem of viscous Burgers' with Dirichlet boundary conditions in Table 10. See Appendices F.1- F.3.
**Periodic boundary condition:** We provide the periodic boundary condition results for Burgers' from Li et al. (2020a) in Appendix F.2.
**Resolution independence for Neumann BC:** In Appendix F.4, we show that BOON trained at lower resolutions for the heat equation (eq. (7)), e.g. $N = 256$, can predict the output at finer resolutions, e.g. $N = 512, 1024$ and $2048$ (See Table 14).
**BOON applied to MWT:** In Appendix F.5, we show that BOON can be successfully applied to other state-of-the-art kernel-based operators, e.g. the multiwavelet-based neural operator (MWT) (Gupta et al., 2021).

## 5 CONCLUSION

In this work, we propose BOON which explicitly utilizes prescribed boundary conditions in a given neural operators' structure by making systematic changes to the operator kernel to enforce their satisfaction. We propose a refinement procedure to enforce common BCs, e.g. Dirichlet, Neumann, and periodic. We also provide error estimates of the procedure. We show empirically, over a range of physically-informative PDE settings, that the proposed approach ensures the satisfaction of the boundary conditions, and leads to better overall accuracy of the solution in the interior of the domain.

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

The appendix is organized as follows. Section A summarizes all the notations used in this work. Section B provides proof of Proposition listed in Section 3.1. The proofs of main Theorems are provided in Section C. The details of data generation process is provided in Section D. Section E further expands the experimental setup in addition to what described in Section 4. Finally, additional experiments to supplement the main findings in Section 4 of the paper are provided in Section F.

## A    NOTATIONS

**PDEs, Boundary value problem (BVP) and operators**

| | |
|---|---|
| $u_0(x)$ | Initial condition of the given PDE. |
| $\mathcal{A}$ | Function space of initial conditions. |
| $u(x, t)$ | Solution of the PDE at time $t > 0$. |
| $\mathcal{U}$ | Function space of solutions of a given PDE. |
| $L^p$ | Function spaces such that the absolute value of the $p$-th power of the element functions is Lebesgue integrable. |
| $\mathcal{H}^{s,p}$ | Sobolev spaces that are subspaces of $L^p$ such that the element functions and their weak derivatives up to order $s > 0$ also belong to $L^p$. |

**Boundary constrained operators**

| | |
|---|---|
| $\partial\mathcal{U}$ | Function space such that each element function satisfies a given boundary condition. |
| $\mathcal{U}_{\text{bdy}}$ | Function space such that each element function is a solution to given a BVP, and satisfies a given boundary condition, i.e. $\mathcal{U}_{\text{bdy}} = \mathcal{U} \cap \partial\mathcal{U}$. |
| $\lVert . \rVert_{\partial\mathcal{H}^{s,p}}$ | $L^p$-norm computed at the boundary points. |
| $\mathcal{E}_{\text{bdy}}(T)$ | Boundary error incurred by a given operator $T$. |

**BOON kernel corrections**

| | |
|---|---|
| $\mathbf{K}$ | Kernel matrix $\mathbf{K} \in \mathbb{R}^{N \times N}$ for a given resolution $N > 0$. |
| $\mathcal{K}$ | Kernel matrix-vector multiplication computing module $\mathbf{y} = \mathcal{K}(\mathbf{x}) := \mathbf{K}\mathbf{x}$. |
| $\mathbf{K}_{\text{bdy}}$ | Boundary corrected Kernel matrix $\mathbf{K}_{\text{bdy}} \in \mathbb{R}^{N \times N}$ for a given resolution $N > 0$. $\mathbf{K}_{\text{bdy}}$ will differ for different boundary conditions. |

## B    PROOF OF PROPOSITION 1

*Proof.* Dirichlet: Given a boundary location $x_0 \in \partial\Omega$, a boundary value function $\alpha_{\text{D}}(x_0, t)$, an input $u_0(x)$ satisfying $u_0(x_0) = \alpha_{\text{D}}(x_0, 0)$, we obtain $u(x_0, t)$ using the integral kernel in eq. (2) with $a(x) := u_0(x)$ as:

$$
\begin{aligned}
u(x_0, t) = T\, u_0(x_0) &= \int_\Omega K(x_0, y) u_0(y) dy \\
&= \int_\Omega \frac{\alpha_{\text{D}}(t)}{\alpha_{\text{D}}(0)} \mathbb{1}(y = x_0) u_0(y) dy \\
&= \frac{\alpha_{\text{D}}(x_0, t)}{\alpha_{\text{D}}(x_0, 0)} u_0(x_0) = \alpha_{\text{D}}(x_0, t).
\end{aligned}
$$

For the corner case of $\alpha_D(x_0, 0) = 0$, the division is not defined. In such cases, we can still obtain the boundary corrected solution with an arbitrary precision $\epsilon > 0$. First, we note that, the operator is

such that $T : \mathcal{A} \to \mathcal{U}$, where $\mathcal{A}, \mathcal{U}$ are $\mathcal{H}^{s,2}$. Therefore, the operator is compact, or it maps bounded norm subspaces.

The operator kernel is $K : \Omega \times \Omega \to L^2$. We first define $\sup_x ||K(x,.)||_{L^2} = M$. Next, let $u_{0\,\epsilon'}(x)$ be the $\epsilon'$-perturbed version of a given input $u_0(x)$ such that $||u_{0\,\epsilon'} - u_0||_{L^2} \leq \epsilon'$ and $u_{0\,\epsilon'}(x_0) = \epsilon'$, where $\epsilon' > 0$. By taking the kernel $K$ such that $K(x_0, y) = \frac{\alpha_D(x_t,t)}{\epsilon'}\mathbb{1}(y = x_0)$, we obtain an $\epsilon$-perturbed output $u_\epsilon(x), \epsilon > 0$ such that

$$u_\epsilon(x_0) = \int_\Omega \frac{\alpha_D(x_0,t)}{\epsilon'}\mathbb{1}(y = x_0)u_{0,\epsilon'}(y)dy = \alpha_D(x_0,t).$$

For any $x \neq x_0$, we denote $u(x)$ as the solution obtained by original input $u_0(x)$, and

$$|u(x) - u_{\epsilon'}(x)| = |\int_\Omega K(x,y)(u_0(y) - u_{0,\epsilon'}(y))dy|$$

$$\overset{(a)}{\leq} ||K(x,.)||_{L^2}||u_0 - u_{0,\epsilon'}||_{L^2}$$

$$\leq \sup_x ||K(x,.)||_{L^2}||u_0 - u_{0,\epsilon'}||_{L^2}$$

$$\leq M\epsilon',$$

where, $(a)$ uses Hölder's inequality. Given an output-approximation $\epsilon$, we take $\epsilon' = \epsilon/M$. Therefore, $\sup_x |u(x) - u_\epsilon(x)| = \epsilon$, or we obtain solution for any given precision $\epsilon$ such that $u_\epsilon(x_0) = \alpha_D(x_0,t)$.

Note that, for the computational purpose, when we evaluate the functions and kernels on a discretized grid (see Appendix C.2.1), we simply replace the appropriate locations with the boundary values, and the modified kernel takes care of the continuity over a given input resolution.

Neumann: Given a boundary location $x_0 \in \partial\Omega$, a boundary function $\alpha_N(x_0,t)$, and an input $u_0(x)$, we obtain the derivative of the output $u_x(x_0,t)$ by differentiating the integral kernel in eq. (2) with $a(x) = u_0(x)$ as:

$$u_x(x_0,t) = \frac{\partial}{\partial x}T\,u_0(x)\Big|_{x=x_0}$$

$$= \int_\Omega \frac{\partial}{\partial x}K(x,y)u_0(y)dy\Big|_{x=x_0}$$

$$= \int_\Omega K_x(x_0,y)u_0(y)dy$$

$$= \int_\Omega \frac{\alpha_N(x_0,t)}{u_0(x_0)}\mathbb{1}(y = x_0)u_0(y)dy$$

$$= \frac{\alpha_N(x_0,t)}{u_0(x_0)}u_0(x_0) = \alpha_N(x_0,t).$$

Note that, a similar $\epsilon, \epsilon'$ argument can be provided to deal with the special case of $u_0(x_0) = 0$ as we did above for the Dirichlet BC.

Periodic: Given boundary locations $x_0, x_{N-1} \in \partial\Omega$, and an input $u_0(x)$ satisfying periodicity, i.e. $u_0(x_0) = u_0(x_{N-1})$, we obtain $u(x_0,t)$ using the integral kernel in eq. (2) with $a(x) = u_0(x)$ as:

$$u(x_0,t) = T\,u_0(x_0) = \int_\Omega K(x_0,y)u_0(y)dy$$

$$= \int_\Omega K(x_{N-1},y)u_0(y)dy$$

$$= T\,u_0(x_{N-1}) = u(x_{N-1},t).$$

$\square$

## C  PROOF OF THE THEOREMS

### C.1  REFINEMENT THEOREM

We first present the supporting Lemmas in Section C.1.1 and then the proof in Section C.1.2.

#### C.1.1  KERNEL CORRECTION LEMMAS

We have a given resolution $N$, finite grid $x_0, x_1, \ldots, x_{N-1}$, and discretized version of the continuous kernel $K(x, y)$ as $\mathbf{K} = [K_{i,j}]_{0:N-1,0:N-1}$ such that $\mathbf{K} \in \mathbb{R}^{N \times N}$. The operation in eq. (2) is discretized as $\mathbf{u}(t) = \mathbf{K}\mathbf{u}_0$, where $\mathbf{u}_0 \in \mathbb{R}^N$ denotes the values of the initial condition evaluated at the gridpoints, and $\mathbf{u}(t) \in \mathbb{R}^N$ denotes the approximated output at the gridpoints for some $t > 0$.

**Lemma 1.** *Given a Dirichlet boundary condition $u(x_0, t) = \alpha_D(x_0, t)$, let $\beta = \frac{\alpha_D(x_0,t)}{\alpha_D(x_0,0)}$, the operator kernel can be modified for satisfying the boundary condition as follows:*

$$\mathbf{K}_{bdy} = \mathbf{T}^1_{Dirichlet}\mathbf{K}\mathbf{T}^2_{Dirichlet}, \tag{9}$$

*where,*

$$\mathbf{T}^1_{Dirichlet} = \begin{bmatrix} \frac{\beta}{K_{0,0}} & \mathbf{0}^T \\ \mathbf{0} & \mathbf{I}_{N-1} \end{bmatrix}, \mathbf{T}^2_{Dirichlet} = \begin{bmatrix} 1 & -\frac{\mathbf{K}^T_{0,1:N-1}}{K_{0,0}} \\ \mathbf{0} & \mathbf{I}_{N-1} \end{bmatrix}. \tag{10}$$

*Proof.* The $\mathbf{K}_{\text{bdy}}$ is computed as:

$$\mathbf{K}_{\text{bdy}} = \begin{bmatrix} \frac{\beta}{K_{0,0}} & \mathbf{0}^T \\ \mathbf{0} & \mathbf{I}_{N-1} \end{bmatrix} \begin{bmatrix} K_{0,0} & \mathbf{K}^T_{0,1:N-1} \\ \mathbf{K}_{1:N-1,0} & \mathbf{K}_{1:N-1,1:N-1} \end{bmatrix} \begin{bmatrix} 1 & -\frac{\mathbf{K}^T_{0,1:N-1}}{K_{0,0}} \\ \mathbf{0} & \mathbf{I}_{N-1} \end{bmatrix}$$

$$= \begin{bmatrix} \beta & \mathbf{0}^T \\ \mathbf{K}_{1:N-1,0} & \mathbf{K}_{1:N-1,1:N-1} - \frac{1}{K_{0,0}}\mathbf{K}_{1:N-1,0}\mathbf{K}^T_{0,1:N-1} \end{bmatrix}.$$

With $\mathbf{u}_0[0] = \alpha_D(x_0, 0)$, the output using modified kernel is $\mathbf{u}(t) = \mathbf{K}_{\text{bdy}}\mathbf{u}_0$ such that $\mathbf{u}(t)[0] = \beta\mathbf{u}_0[0] = \alpha_D(x_0, t)$. □

**Lemma 2.** *The continuous Neumann boundary condition $u_x(x_0, t) = \alpha_N(x_0, t)$ after discretization through a general one-sided finite difference scheme, $u_x(x_0, t) \approx \sum_{k>0} c_k\mathbf{u}[k](t) + c_0\mathbf{u}[0](t)$ for given coefficients $c_k \in \mathbb{R}$ for $k > 0$ and $c_0 \in \mathbb{R}\backslash\{0\}$, and the following condition for discretized kernel*

$$K_{0,j} = \begin{cases} -\sum_{k>0} \frac{c_k}{c_0} K_{k,j}, & j \neq 0, \\ -\sum_{k>0} \frac{c_k}{c_0} K_{k,j} + \frac{\alpha_N(t)}{c_0 \mathbf{u}_0[0]}, & j = 0, \end{cases} \tag{11}$$

*are sufficient for a $\mathbf{K}$ to satisfy discretized Neumann boundary condition.*

*Proof.* We show sufficiency through the discretized Neumann boundary condition, $u_x(x_0, t) \approx \sum_{k>0} c_k\mathbf{u}[k](t) + c_0\mathbf{u}[0](t)$ as

$$u_x(x_0, t) \approx \sum_{k>0} c_k\mathbf{u}[k](t) + c_0\mathbf{u}[0](t)$$

$$= \sum_{k>0} c_k(\mathbf{K}\mathbf{u}_0)[k] + c_0(\mathbf{K}\mathbf{u}_0)[0]$$

$$= \sum_{k>0} \sum_{j=0}^{N-1} c_k K_{k,j}\mathbf{u}_0[j] + c_0 \sum_{j=0}^{N-1} K_{0,j}\mathbf{u}_0[j]$$

$$= \sum_{j=0}^{N-1} c_0 \left( \sum_{k>0} \frac{c_k}{c_0} K_{k,j} + K_{0,j} \right) \mathbf{u}_0[j]$$

$$= c_0 \frac{\alpha_N(x_0, t)}{c_0 \mathbf{u}_0[0]}\mathbf{u}_0[0] = \alpha_N(x_0, t),$$

where we used eq. (11) to obtain the last equality. □

**Lemma 3.** *Given a Neumann boundary condition $u_x(x_0, t) = \alpha_N(x_0, t)$ and its discretization through a general one-sided finite difference scheme, $u_x(x_0, t) \approx \sum_{k>0} c_k \mathbf{u}[k](t) + c_0 \mathbf{u}[0](t)$ for given coefficients $c_k \in \mathbb{R}$, $c_0 \neq 0$, and $\beta = \frac{\alpha_N(x_0,t)}{c_0\,u_0(x_0)}$, the operator kernel can be modified through following operations to satisfy discretized Neumann boundary conditions.*

$$\mathbf{K}_{bdy} = \mathbf{T}^1_{Neumann}\mathbf{K}\mathbf{T}^2_{Neumann}, \tag{12}$$

*where,*

$$\mathbf{T}^1_{Neumann} = \begin{bmatrix} \frac{\beta}{K_{0,0}} & -\frac{c_1}{c_0} & \cdots & -\frac{c_{N-1}}{c_0} \\ \mathbf{0} & & \mathbf{I}_{N-1} & \end{bmatrix}, \mathbf{T}^2_{Neumann} = \begin{bmatrix} 1 & -\frac{\mathbf{K}^T_{0,1:N-1}}{K_{0,0}} \\ \mathbf{0} & \mathbf{I}_{N-1} \end{bmatrix}.$$

*Proof.* The $\mathbf{K}_{\text{bdy}}$ is evaluated as

$$
\mathbf{K}_{\text{bdy}} = \begin{bmatrix} \frac{\beta}{K_{0,0}} & -\frac{c_1}{c_0} & \cdots & -\frac{c_{N-1}}{c_0} \\ \mathbf{0} & & \mathbf{I}_{N-1} & \end{bmatrix} \begin{bmatrix} K_{0,0} & \mathbf{K}^T_{0,1:N-1} \\ \mathbf{K}_{1:N-1,0} & \mathbf{K}_{1:N-1,1:N-1} \end{bmatrix} \begin{bmatrix} 1 & -\frac{1}{K_{0,0}}\mathbf{K}^T_{0,1:N-1} \\ \mathbf{0} & \mathbf{I}_{N-1} \end{bmatrix}
$$

$$
= \begin{bmatrix} \beta - \frac{1}{c_0}\sum_{k>0} c_k K_{k,0} & -\frac{1}{c_0}\sum_{k>0} c_k(\mathbf{K}^T_{k,1:N-1} - \frac{1}{K_{0,0}}K_{k,0}\mathbf{K}^T_{0,1:N-1}) \\ \mathbf{K}_{1:N-1,0} & \mathbf{K}_{1:N-1,1:N-1} - \frac{1}{K_{0,0}}\mathbf{K}_{1:N-1,0}\mathbf{K}^T_{0,1:N-1} \end{bmatrix}.
$$

Now, $(K_{\text{bdy}})_{i,0} = K_{i,0}, \forall\, i > 0$, therefore, (i) $(K_{\text{bdy}})_{0,0} = \beta - \frac{1}{c_0}\sum_{k>0} c_k(K_{\text{bdy}})_{k,0}$. Next, $(K_{\text{bdy}})_{i,j} = K_{i,j} - \frac{1}{K_{0,0}}K_{i,0}K_{0,j} \,\forall\, i, j > 0$, therefore, (ii) $(K_{\text{bdy}})_{0,j} = \sum_{k>0} \frac{c_k}{c_0}(K_{\text{bdy}})_{k,j}$. Using Lemma 2, $\mathbf{K}_{\text{bdy}}$ satisfies discretized Neumann boundary condition. $\qquad\square$

**Lemma 4.** *Given a periodic boundary condition $\mathbf{u}[0](t) = \mathbf{u}[N-1](t)\,\forall\, t$, the operator kernel can be modified to satisfy the periodicity as follows.*

$$\mathbf{K}_{bdy} = \mathbf{T}_{periodic}\mathbf{K}, \tag{13}$$

*where,*

$$\mathbf{T}_{periodic} = \begin{bmatrix} \alpha & \mathbf{0}^T & \beta \\ \mathbf{0} & \mathbf{I}_{N-2} & \mathbf{0} \\ \alpha & \mathbf{0}^T & \beta \end{bmatrix}, \quad \alpha, \beta \in \mathbb{R}^+, \alpha + \beta = 1. \tag{14}$$

*Proof.* The $\mathbf{K}_{\text{bdy}}$ is computed as

$$
\begin{aligned}
\mathbf{K}_{\text{bdy}} &= \mathbf{T}_{\text{periodic}}\mathbf{K} \\
&= \begin{bmatrix} \alpha & \mathbf{0}^T & \beta \\ \mathbf{0} & \mathbf{I}_{N-2} & \mathbf{0} \\ \alpha & \mathbf{0}^T & \beta \end{bmatrix} \begin{bmatrix} \mathbf{K}^T_{0,0:N-1} \\ \mathbf{K}_{1:N-2,0:N-1} \\ \mathbf{K}^T_{N-1,0:N-1} \end{bmatrix} \\
&= \begin{bmatrix} \alpha\,\mathbf{K}^T_{0,0:N-1} + \beta\,\mathbf{K}^T_{N-1,0:N-1} \\ \mathbf{K}_{1:N-2,0:N-1} \\ \alpha\,\mathbf{K}^T_{0,0:N-1} + \beta\,\mathbf{K}^T_{N-1,0:N-1} \end{bmatrix}.
\end{aligned} \tag{15}
$$

With periodic input, i.e., $\mathbf{u}_0[0] = \mathbf{u}_0[N-1]$, the output through modified kernel in eq. (15) or $\mathbf{u}(t) = \mathbf{K}_{\text{bdy}}\mathbf{u}_0$ is such that $\mathbf{u}[0](t) = \mathbf{u}[N-1](t)$. $\qquad\square$

### C.1.2 PROOF OF THEOREM 1

*Proof.* Lemma 1 and Lemma 4 ensure sufficient condition for Dirichlet and periodic boundary conditions, respectively. An approximate satisfaction, through one-sided discretization of derivative, is shown in Lemma 3 for Neumann boundary condition. Therefore, Theorem 1 holds. $\qquad\square$

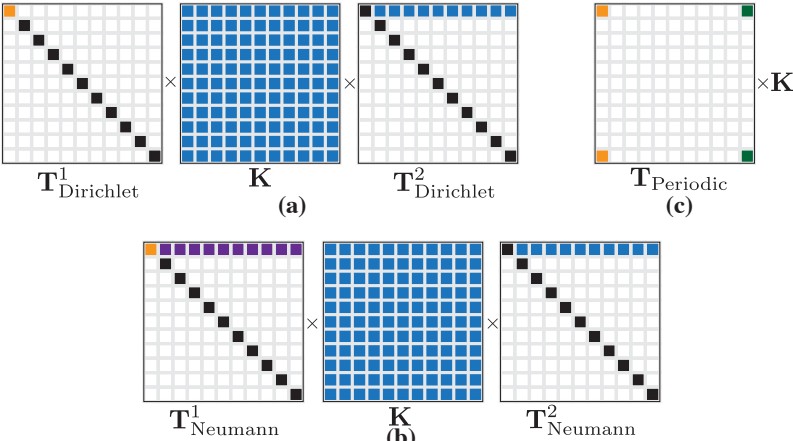

Figure 5: **Operator refinement procedure.** Kernel correction for Dirichlet, Neumann, and periodic in **(a)**, **(b)**, and **(c)**, respectively. The purple color in **(b)** represents pre-computed weights for a first-order derivative approximation (see Lemma 3).

## C.2   COMPLEXITY THEOREM

Given a resolution $N$, the kernel matrix $\mathbf{K} \in \mathbb{R}^{N \times N}$ is, in-general, dense. Therefore, for an input $\mathbf{u}_0$, the operation $\mathbf{u} = \mathbf{K}_{\text{bdy}}(\mathbf{u}_0)$ using the proposed kernel corrections in Lemma 1, 3, and 4 have a cost complexity of $O(N^3)$, and space complexity of $O(N^2)$ through a naive implementation. We show that if an efficient implementation exists for the matrix-vector operation $\mathbf{y} = \mathbf{K}\mathbf{x}$, say $\mathbf{y} := \mathcal{K}(\mathbf{x})$ which has cost complexity $O(N_{\mathcal{K}})$ (for example, Fourier bases in Li et al. (2020a), multiwavelets in Gupta et al. (2021)), then an efficient implementation exists for our kernel correction (BOON) as we show in the following sections.

### C.2.1   DIRICHLET

To implement the correction operation of Lemma 1, we proceed with eq. (9) as follows.

$$\mathbf{u} = \mathbf{T}^1_{\text{Dirichlet}} \mathbf{K} \mathbf{T}^2_{\text{Dirichlet}} \mathbf{u}_0. \tag{16}$$

First, let $\tilde{\mathbf{y}} = \mathbf{T}^2_{\text{Dirichlet}} \mathbf{u}_0$, therefore

$$\tilde{\mathbf{y}} = \begin{bmatrix} \mathbf{u}_0[0] - \frac{1}{K_{0,0}} \mathbf{K}^T_{0,1:N-1} \mathbf{u}_0[1:N-1] \\ \mathbf{u}_0[1] \\ \vdots \\ \mathbf{u}_0[N-1] \end{bmatrix} \tag{17}$$

$$= \begin{bmatrix} 2\mathbf{u}_0[0] - \frac{1}{K_{0,0}} \mathbf{K}^T_{0,0:N-1} \mathbf{u}_0 \\ \mathbf{u}_0[1] \\ \vdots \\ \mathbf{u}_0[N-1] \end{bmatrix} \tag{18}$$

$$= \begin{bmatrix} 2\mathbf{u}_0[0] - \frac{1}{(\mathbf{K}\mathbf{e}_0)[0]} (\mathbf{K}\mathbf{u}_0)[0] \\ \mathbf{u}_0[1] \\ \vdots \\ \mathbf{u}_0[N-1] \end{bmatrix}, \tag{19}$$

where $\mathbf{e}_0 = [1, 0, \ldots, 0]^T$ is the first eigenvector, therefore, $K_{0,0} = (\mathbf{K}\mathbf{e}_0)[0]$. Now, if $\mathbf{z} = \mathbf{K}\mathbf{u}_0$, then $\tilde{\mathbf{y}}$ in eq. (19) is written as

$$\tilde{\mathbf{y}} = \begin{bmatrix} 2\mathbf{u}_0[0] - \frac{1}{(\mathbf{K}\mathbf{e}_0)[0]}\mathbf{z}[0] \\ \mathbf{u}_0[1] \\ \vdots \\ \mathbf{u}_0[N-1] \end{bmatrix}. \tag{20}$$

Next, let $\hat{\mathbf{y}} = \mathbf{K}\mathbf{T}_{\text{Dirichlet}}^2\mathbf{u}_0$, or $\hat{\mathbf{y}} = \mathbf{K}\tilde{\mathbf{y}}$. Finally, we denote $\mathbf{u} = \mathbf{T}_{\text{Dirichlet}}^1\hat{\mathbf{y}}$, or

$$\mathbf{u} = \begin{bmatrix} \frac{\alpha_D(x_0,t)}{\alpha_D(x_0,0)} \frac{1}{(\mathbf{K}\mathbf{e}_0)[0]}\hat{\mathbf{y}}[0] \\ \hat{\mathbf{y}}[1] \\ \vdots \\ \hat{\mathbf{y}}[N-1] \end{bmatrix} \tag{21}$$

$$= \begin{bmatrix} \frac{\alpha_D(x_0,t)}{\mathbf{u}_0[0]\,(\mathbf{K}\mathbf{e}_0)[0]}\hat{\mathbf{y}}[0] \\ \hat{\mathbf{y}}[1] \\ \vdots \\ \hat{\mathbf{y}}[N-1] \end{bmatrix}, \tag{22}$$

Further, we observe that

$$\hat{\mathbf{y}}[0] = \mathbf{K}_{0,0:N-1}^T\tilde{\mathbf{y}}$$
$$= K_{0,0}(\mathbf{u}_0[0] - \frac{1}{K_{0,0}}\mathbf{K}_{0,1:N-1}^T\mathbf{u}_0[1:N-1]) + \mathbf{K}_{0,1:N-1}^T\mathbf{u}_0[1:N-1] \tag{23}$$
$$= K_{0,0}\mathbf{u}_0[0].$$

We finally express $\mathbf{u}$ as

$$\mathbf{u} = \begin{bmatrix} \alpha_D(x_0,t) \\ \hat{\mathbf{y}}[1] \\ \vdots \\ \hat{\mathbf{y}}[N-1] \end{bmatrix}, \tag{24}$$

**Note:** For the above kernel correction implementation, we never require an explicit knowledge of the kernel matrix $\mathbf{K}$ but only a matrix-vector multiplication module $\mathbf{y} = \mathcal{K}(\mathbf{x}) := \mathbf{K}\mathbf{x}$. Therefore, the above operations for Dirichet correction can be implemented with 3 kernel $\mathcal{K}$ calls and $O(N)$ space complexity. The steps are summarized in the Section 3.2 of main text as Algorithm 1.

**Remark:** For the special case when $\alpha_D(x_0, 0) = 0$, we note that $\beta$ in Lemma 1 is not defined but due to eq.(23), we can omit the division and still obtain boundary satisfied output through eq.(24).

### C.2.2   NEUMANN

The correction using Lemma 3 is implemented via eq. (12) as follows.

$$\mathbf{u} = \mathbf{T}_{\text{Neumann}}^1\mathbf{K}\mathbf{T}_{\text{Neumann}}^2\mathbf{u}_0. \tag{25}$$

Since $\mathbf{T}_{\text{Neumann}}^2 = \mathbf{T}_{\text{Dirichlet}}^2$, therefore, we follow the same initial steps as outlined in Section C.2.1 to first obtain $\tilde{\mathbf{y}} = \mathbf{T}_{\text{Neumann}}^2\mathbf{u}_0$ as

$$\tilde{\mathbf{y}} = \begin{bmatrix} 2\mathbf{u}_0[0] - \frac{1}{(\mathbf{K}\mathbf{e}_0)[0]}\mathbf{z}[0] \\ \mathbf{u}_0[1] \\ \vdots \\ \mathbf{u}_0[N-1] \end{bmatrix}, \tag{26}$$

where, $K_{0,0} = (\mathbf{K}\mathbf{e}_0)[0]$ and $\mathbf{z} = \mathbf{K}\mathbf{u}_0$, as mentioned in Section C.2.1. Next, let $\hat{\mathbf{y}} = \mathbf{K}\tilde{\mathbf{y}}$ and denote $\mathbf{u} = \mathbf{T}_{\text{Neumann}}^1\hat{\mathbf{y}}$, or

$$\mathbf{u} = \begin{bmatrix} \frac{\beta}{K_{0,0}} & -\frac{c_1}{c_0} & \cdots & -\frac{c_{N-1}}{c_0} \\ \mathbf{0} & & \mathbf{I}_{N-1} & \end{bmatrix} \hat{\mathbf{y}} \tag{27}$$

$$
= \begin{bmatrix} \frac{\beta}{(\mathbf{Ke}_0)[0]} \hat{\mathbf{y}}[0] - \sum_{k>0} \frac{c_k}{c_0} \hat{\mathbf{y}}[k] \\ \hat{\mathbf{y}}[1] \\ \vdots \\ \hat{\mathbf{y}}[N-1] \end{bmatrix} \tag{28}
$$

$$
= \begin{bmatrix} \frac{\alpha_{\mathrm{N}}(x_0,t)}{c_0 \, (\mathbf{Ke}_0)[0] \mathbf{u}_0[0]} \hat{\mathbf{y}}[0] - \sum_{k>0} \frac{c_k}{c_0} \hat{\mathbf{y}}[k] \\ \hat{\mathbf{y}}[1] \\ \vdots \\ \hat{\mathbf{y}}[N-1] \end{bmatrix}, \tag{29}
$$

where the last equality uses $\beta$ from Lemma 3. Similar to Section C.2.1, we observe that

$$
\begin{aligned}
\hat{\mathbf{y}}[0] &= \mathbf{K}_{0,0:N-1}^T \tilde{\mathbf{y}} \\
&= K_{0,0}(\mathbf{u}_0[0] - \frac{1}{K_{0,0}} \mathbf{K}_{0,1:N-1}^T \mathbf{u}_0[1:N-1]) + \mathbf{K}_{0,1:N-1}^T \mathbf{u}_0[1:N-1] \\
&= K_{0,0} \mathbf{u}_0[0]
\end{aligned} \tag{30}
$$

Therefore, we finally express $\mathbf{u}$ as

$$
\mathbf{u} = \begin{bmatrix} \frac{\alpha_{\mathrm{N}}(x_0,t)}{c_0} \hat{\mathbf{y}}[0] - \sum_{k>0} \frac{c_k}{c_0} \hat{\mathbf{y}}[k] \\ \hat{\mathbf{y}}[1] \\ \vdots \\ \hat{\mathbf{y}}[N-1] \end{bmatrix} \tag{31}
$$

**Note:** Similar to Section C.2.1, the Neumann kernel corrections also do not require an explicit knowledge of the kernel matrix and only a matrix-vector multiplication module $\mathbf{y} = \mathbf{Kx} := \mathcal{K}(\mathbf{x})$. Three calls to the kernel module $\mathcal{K}$ are sufficient and can be implemented in $O(N)$ space with finite-difference coefficients $c_0 \neq 0, c_k \, k > 0$ as an input. The steps are outlined in the Section 3.2 as Algorithm 2.

**Remark:** For the special case when $\mathbf{u}_0[0] = 0$, we note that $\beta$ in Lemma 3 is not defined. But due to eq.(30), we can omit the division and still obtain boundary satisfied output through eq.(31).

### C.2.3 PERIODIC

The periodic kernel correction is implemented using Lemma 4 and eq.(13) as follows.

$$
\mathbf{u} = \mathbf{T}_{\mathrm{Periodic}} \mathbf{Ku}_0. \tag{32}
$$

Let $\hat{y} = \mathbf{Ku}_0$, therefore, $\mathbf{u}$ is written as $\mathbf{u} = \mathbf{T}_{\mathrm{Periodic}} \hat{\mathbf{y}}$, or

$$
\mathbf{u} = \mathbf{T}_{\mathrm{Periodic}} \hat{\mathbf{y}} \tag{33}
$$

$$
= \begin{bmatrix} \alpha & \mathbf{0}^T & \beta \\ \mathbf{0} & \mathbf{I}_{N-2} & \mathbf{0} \\ \alpha & \mathbf{0}^T & \beta \end{bmatrix} \hat{\mathbf{y}} \tag{34}
$$

$$
= \begin{bmatrix} \alpha \hat{\mathbf{y}}[0] + \beta \hat{\mathbf{y}}[N-1] \\ \hat{\mathbf{y}}[1] \\ \vdots \\ \hat{\mathbf{y}}[N-2] \\ \alpha \hat{\mathbf{y}}[0] + \beta \hat{\mathbf{y}}[N-1] \end{bmatrix}. \tag{35}
$$

The steps are outlined as Algorithm 3 in Section 3.2. Note that, periodic kernel correction only requires one matrix-vector multiplication which can be obtained via $\mathbf{y} = \mathbf{Kx} := \mathcal{K}(\mathbf{x})$.

### C.3 KERNEL CORRECTION ARCHITECTURE

The neural architectures of BOON using Algorithm 1,2, and 3 for Dirichlet, Neumann, and Periodic, respectively, are shown in Figure 6.

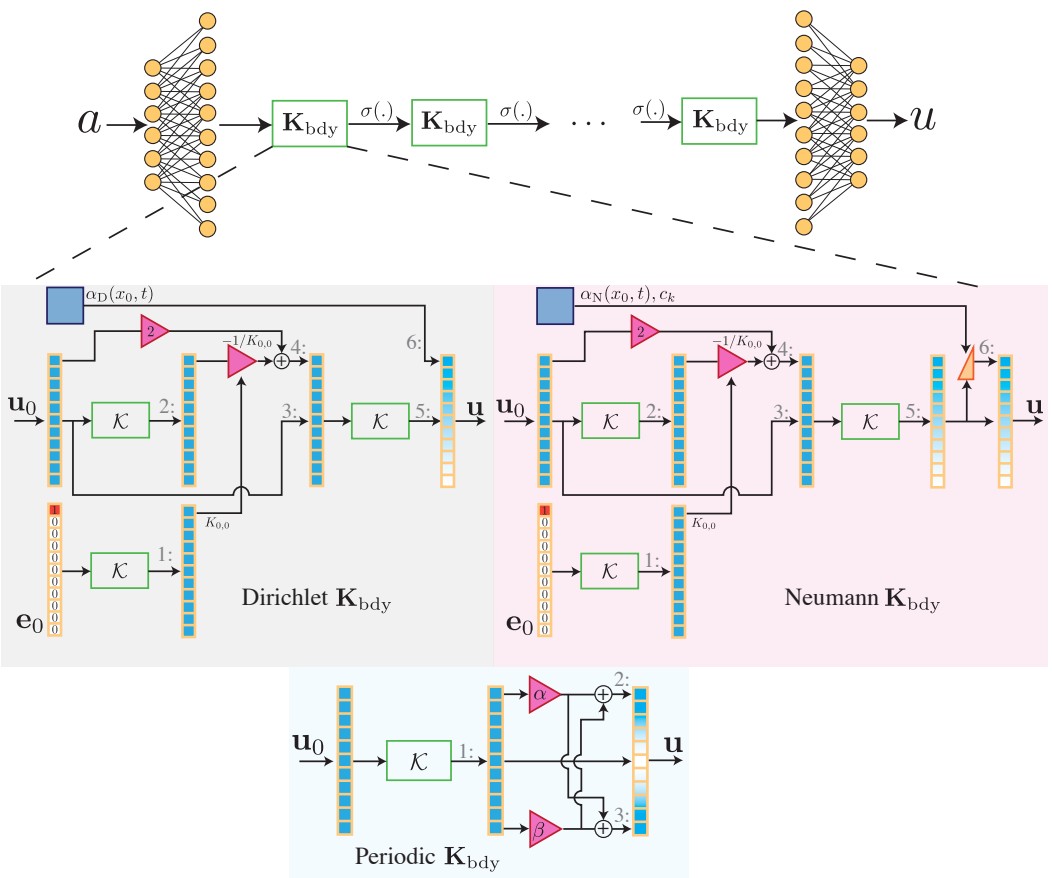

Figure 6: **BOON architectures.** Kernel correction architectures for Dirichlet, Neumann, and periodic using Algorithm 1, 2, and 3, respectively. The line numbers of the algorithms are shown at the corresponding locations. The complete BOON uses multiple corrected kernels $\mathbf{K}_{\text{bdy}}$ in concatenation along with non-linearity $\sigma(.)$. Each correction stage uses a kernel computing module $\mathcal{K}$ from the base neural operator, for example, FNO (Li et al., 2020a), MWT (Gupta et al., 2021). The multiple appearances of $\mathcal{K}$ within each boundary correction architecture share the same parameters $\theta_{\mathcal{K}}$. The MLP layers at the beginning and end are used to upscale and downscale, respectively, the channel dimensions for better learning. See Section 4 for more details.

## C.4 BOUNDEDNESS THEOREM

### C.4.1 PERIODIC

**Lemma 5.** *For a given continuous input $u_0(x) \in L^2$, the output obtained by original operator $T$ and kernel corrected $\tilde{T}$ through Proposition 1 for periodic are $u(x,t)$ and $\tilde{u}(x,t)$, respectively. The new kernel $\tilde{K}(x,y)$ is such that*

$$\tilde{K}(x,y) = \begin{cases} \alpha\, K(x_0,y) + \beta\, K(x_{N-1},y) & x = x_0, x_0 \leq y \leq x_{N-1}, \\ \alpha\, K(x_0,y) + \beta\, K(x_{N-1},y) & x = x_{N-1}, x_0 \leq y \leq x_{N-1}, \end{cases} \tag{36}$$

*where $\alpha, \beta \in \mathbb{R}^+$, $\alpha + \beta = 1$. The solutions are bounded in $L^2$-norm as follows.*

$$\|u(x,t) - \tilde{u}(x,t)\|_{L^2} = \sqrt{\left[(1-\alpha)^2 + (1-\beta)^2\right]} \cdot |u(x_0,t) - u(x_{N-1},t)|. \tag{37}$$

*Additionally, if $\alpha = \beta = 0.5$, then the upper bound in (37) is minimized and we have,*

$$\|u(x,t) - \tilde{u}(x,t)\|_{L^2} \leq \frac{1}{2}|u(x_0,t) - u(x_{N-1},t)|. \tag{38}$$

*Proof.* We take the continuous input domain $\Omega = [x_0, x_{N-1}]$, the difference between the solutions is written as.

$$\|u(x,t) - \tilde{u}(x,t)\|_{L_2}^2 = \int_\Omega (u(x,t) - \tilde{u}(x,t))^2 dx$$

$$= \int_\Omega \Big[ \int_\Omega K(x,y) u_0(y) dy - \int_\Omega \tilde{K}(x,y) u_0(y) dy \Big]^2 dx$$

$$= \int_\Omega \Big[ \int_\Omega \big[ K(x,y) - \tilde{K}(x,y) \big] u_0(y) dy \Big]^2 dx.$$

Moreover, from Proposition 1, the kernels differ as

$$K(x,y) - \tilde{K}(x,y) = \begin{cases} 0, & x_0 < x < x_{N-1}, x_0 \le y \le x_{N-1}, \\ (1-\alpha)K(x_0, y) - \beta K(x_{N-1}, y), & x = x_0, x_0 \le y \le x_{N-1}, \\ -\alpha K(x_0, y) + (1-\beta)K(x_{N-1}, y), & x = x_{N-1}, x_0 \le y \le x_{N-1}. \end{cases} \tag{39}$$

Therefore, using eq. (39), we have

$$\|u(x,t) - \tilde{u}(x,t)\|_{L^2}^2 = (u(x_0,t) - \tilde{u}(x_0,t))^2 + (u(x_{N-1},t) - \tilde{u}(x_{N-1},t))^2$$

$$= \Big( (1-\alpha)u(x_0,t) - \beta u(x_{N-1},t) \Big)^2 + \Big( -\alpha u(x_0,t) + (1-\beta)u(x_{N-1},t) \Big)^2$$

$$\overset{(a)}{=} (1-\alpha)^2 (u(x_0,t) - u(x_{N-1},t))^2 + (1-\beta)^2 (u(x_0,t) - u(x_{N-1},t))^2$$

$$= \Big[ (1-\alpha)^2 + (1-\beta)^2 \Big] (u(x_0,t) - u(x_{N-1},t))^2.$$

where in $(a)$ we use $\alpha + \beta = 1$. Again, the bound can be further tightened as

$$\|u(x,t) - \tilde{u}(x,t)\|_{L^2} \le \sqrt{1 - 2\alpha + 2\alpha^2} \cdot |u(x_0,t) - u(x_{N-1},t)|, \tag{40}$$

which attains a minimum at $\alpha = 0.5$. $\qquad\square$

**Corollary 1.** *A discretized input $\mathbf{u}_0 \in \mathbb{R}^N$ through discretized kernels $\mathbf{K}$ and $\mathbf{K}_{bdy}$ (from Lemma 4) results in $\mathbf{u}(t)$ and $\tilde{\mathbf{u}}(t)$ such that*

$$\|\mathbf{u}(t) - \tilde{\mathbf{u}}(t)\|_2 \le \frac{1}{2} |\mathbf{u}[0](t) - \tilde{\mathbf{u}}[0](t)|. \tag{41}$$

### C.4.2   DIRICHLET

**Lemma 6.** *For a given continuous input $u_0(x) \in L^2$, the output obtained by original operator $T$ and kernel corrected $\tilde{T}$ through Proposition 1 for Dirichlet are $u(x,t)$ and $\tilde{u}(x,t)$, respectively. We take $C = \frac{\alpha_D(x_0,t)}{\alpha_D(x_0,0)}$, the new kernel is written as*

$$\tilde{K}(x,y) = \begin{cases} C \mathbb{1}(y = x_0) & x = x_0, x_0 \le y \le x_{N-1} \\ K(x,y) & o/w \end{cases}, \tag{42}$$

*The outputs are bounded as*

$$\|u(x,t) - \tilde{u}(x,t)\|_{L_2}$$

$$= \sqrt{\Big[ u(x_0,t) - \alpha_D(x_0,t) \Big]^2 + \Big[ u(x_0,t) - K(x_0,x_0)u_0(x_0) \Big]^2 \Big[ \frac{\int_\Omega K^2(x,x_0)dx}{K^2(x_0,x_0)} - 1 \Big]} \tag{43}$$

*Proof.* The difference between outputs is written as

$$\|u(x,t) - \tilde{u}(x,t)\|_{L^2}^2 = \int_\Omega (u(x,t) - \tilde{u}(x,t))^2 dx$$

$$= \int_\Omega \Big[ \int_\Omega \big[ K(x,y) - \tilde{K}(x,y) \big] u_0(y) dy \Big]^2 dx.$$

The kernels differ as follows

$$K(x,y) - \tilde{K}(x,y) = \begin{cases} K(x_0, x_0) - C & x = x_0, y = x_0 \\ K(x_0, y) & x = x_0, x_0 < y \leq x_{N-1} \\ 0 & o/w \end{cases}. \tag{44}$$

Using eq. (44), we can bound the difference between $u(x_0, t)$ and $\tilde{u}(x_0, t)$ as

$$\|u(x,t) - \tilde{u}(x,t)\|_{L^2}^2 = \left[ \int_\Omega K(x_0, y) u_0(y) dy - C u_0(x_0) \right]^2$$
$$= (u(x_0, t) - \alpha_D(x_0, t))^2.$$

$\square$

**Corollary 2.** *A discretized input* $\mathbf{u}_0 \in \mathbb{R}^N$ *through discretized kernels* $\mathbf{K}$ *and* $\mathbf{K}_{bdy}$ *(from Lemma 1) results in* $\mathbf{u}(t)$ *and* $\tilde{\mathbf{u}}(t)$ *such that*

$$\|\mathbf{u}(t) - \tilde{\mathbf{u}}(t)\|_2 = \sqrt{(\mathbf{u}[0](t) - \alpha_D(x_0, t))^2 + (\mathbf{u}[0](t) - K_{0,0}\mathbf{u}_0[0])^2 \left( \sum_i \frac{K_{i,0}^2}{K_{0,0}^2} - 1 \right)}. \tag{45}$$

*Proof.* Let $C = \alpha_D(x_0, t)/\alpha_D(x_0, 0)$. The difference between the kernels, $\mathbf{K}$ and corrected version $\mathbf{K}_{bdy}$ from Lemma 1 is

$$\mathbf{K}_{i,j} - \mathbf{K}_{bdy\,i,j} = \begin{cases} K_{0,0} - C & i = 0, j = 0 \\ K_{i,j} & i = 0, 1 \leq j \leq N - 1 \\ \frac{1}{K_{0,0}} K_{i,0} K_{0,j} & o/w \end{cases}. \tag{46}$$

Now, estimating the difference between $\mathbf{u}(t)$ and $\tilde{\mathbf{u}}(t)$, we have

$$\mathbf{u}(t) - \tilde{\mathbf{u}}(t) = \mathbf{K}\mathbf{u}_0 - \mathbf{K}_{bdy}\mathbf{u}_0$$
$$= \begin{bmatrix} (K_{0,0} - C)\mathbf{u}_0[0] + \mathbf{K}_{0,1:N-1}^T \mathbf{u}_0[1:N-1] \\ \frac{1}{K_{0,0}} \mathbf{K}_{1:N-1,0} \mathbf{K}_{0,1:N-1}^T \mathbf{u}_0[1:N-1] \end{bmatrix}$$
$$= \begin{bmatrix} \mathbf{u}[0](t) - \alpha_D(t) \\ \frac{1}{K_{0,0}} \mathbf{K}_{1:N-1,0} \mathbf{K}_{0,1:N-1}^T \mathbf{u}_0[1:N-1] \end{bmatrix},$$

Therefore, the norm of difference is written as

$$\|\mathbf{u}(t) - \tilde{\mathbf{u}}(t)\|_2^2 = (\mathbf{u}[0](t) - \alpha_D(x_0, t))^2 + \frac{1}{K_{0,0}^2} \|\mathbf{K}_{1:N-1,0}\mathbf{K}_{0,1:N-1}^T \mathbf{u}_0\|_2^2$$

$$= (\mathbf{u}[0](t) - \alpha_D(x_0, t))^2 + (\sum_{j=1}^{N-1} K_{0,j}\mathbf{u}_0[j])^2 \sum_{i=1}^{N-1} \frac{K_{i,0}^2}{K_{0,0}^2}$$

$$= (\mathbf{u}[0](t) - \alpha_D(x_0, t))^2 + (\mathbf{u}[0](t) - K_{0,0}\mathbf{u}_0[0])^2 \left( \sum_{i=0}^{N-1} \frac{K_{i,0}^2}{K_{0,0}^2} - 1 \right).$$

**Note**: When the kernel is diagonally dominant, i.e., $\sum_{i=0}^{N-1} K_{i,0}^2 \approx K_{0,0}^2$, then the overall difference between the solutions is boundary error. $\square$

### C.4.3 NEUMANN

**Lemma 7.** *A discretized input* $\mathbf{u}_0 \in \mathbb{R}^N$ *through discretized kernels* $\mathbf{K}$ *and* $\mathbf{K}_{bdy}$ *(from Lemma 7) results in* $\mathbf{u}(t)$ *and* $\tilde{\mathbf{u}}(t)$ *such that*

$$\|\mathbf{u}(t) - \tilde{\mathbf{u}}(t)\|_2 \leq \sqrt{f_1^2 + f_2^2}, \tag{47}$$

*where,*

$$f_1^2 = \left( \mathbf{u}[0](t) - \frac{\alpha_N(x_0, t)}{c_0} + \frac{1}{c_0} \sum_{k>0} c_k \mathbf{u}[k](t) + \sum_{k>0} \frac{c_k K_{k,0}}{c_0 K_{0,0}} (K_{0,0}\mathbf{u}_0[0] - \mathbf{u}[0](t)) \right)^2,$$

$$f_2^2 = (\mathbf{u}[0](t) - K_{0,0}\mathbf{u}_0[0])^2 \left( \sum_{i=0}^{N-1} \frac{K_{i,0}^2}{K_{0,0}^2} - 1 \right).$$

*Proof.* Using $\beta = \frac{\alpha_N(x_0,t)}{c_0 \mathbf{u}_0[0]}$, the difference between the kernels, $\mathbf{K}$ and corrected version $\mathbf{K}_{\text{bdy}}$ from Lemma 3 is

$$\mathbf{K}_{i,j} - \mathbf{K}_{\text{bdy}\,i,j} = \begin{cases} K_{0,0} - \beta + \frac{1}{c_0}\sum_{k>0} c_k K_{k,0}, & i=0, j=0, \\ K_{0,j} + \frac{1}{c_0}\sum_{k>0} c_k[K_{k,j} - \frac{1}{K_{0,0}}K_{k,0}K_{0,j}], & i=0, j>0, \\ 0, & i>0, j=0, \\ \frac{1}{K_{0,0}}K_{i,0}K_{0,j}, & \text{o/w.} \end{cases} \tag{48}$$

Therefore, difference between $\mathbf{u}(t)$ and $\tilde{\mathbf{u}}(t)$ is written as

$$\mathbf{u}(t) - \tilde{\mathbf{u}}(t) = (\mathbf{K} - \mathbf{K}_{\text{bdy}})\mathbf{u}_0$$

$$= \begin{bmatrix} (K_{0,0} - \beta + \frac{1}{c_0}\sum_{k>0} c_k K_{k,0})\mathbf{u}_0[0] \\ + \sum_{j>0}(K_{0,j} + \frac{1}{c_0}\sum_{k>0} c_k[K_{k,j} - \frac{1}{K_{0,0}}K_{k,0}K_{0,j}]\mathbf{u}_0[j]) \\ \\ \frac{1}{K_{0,0}}\mathbf{K}_{1:N-1,0}\mathbf{K}_{0,1:N-1}^T\mathbf{u}_0[1:N-1] \end{bmatrix}.$$

Therefore, the norm of difference is written as

$$\|\mathbf{u}(t) - \tilde{\mathbf{u}}(t)\|_2^2 = \left((K_{0,0} - \beta + \frac{1}{c_0}\sum_{k>0} c_k K_{k,0})\mathbf{u}_0[0] + \right.$$

$$\left. \sum_{j>0}(K_{0,j} + \frac{1}{c_0}\sum_{k>0} c_k[K_{k,j} - \frac{1}{K_{0,0}}K_{k,0}K_{0,j}]\mathbf{u}_0[j])\right)^2$$

$$+ \frac{1}{K_{0,0}^2}\|\mathbf{K}_{1:N-1,0}\mathbf{K}_{0,1:N-1}^T\mathbf{u}_0\|_2^2$$

$$= \left(\mathbf{u}[0](t) - \frac{\alpha_N(x_0,t)}{c_0} + \frac{1}{c_0}\sum_{k>0} c_k\mathbf{u}[k](t) + \sum_{k>0}\frac{c_k K_{k,0}}{c_0 K_{0,0}}(K_{0,0}\mathbf{u}_0[0] - \mathbf{u}[0](t))\right)^2$$

$$+ (\mathbf{u}[0](t) - K_{0,0}\mathbf{u}_0[0])^2\left(\sum_{i=0}^{N-1}\frac{K_{i,0}^2}{K_{0,0}^2} - 1\right).$$

$\square$

### C.4.4 PROOF OF THEOREM 3

*Proof.* Using Lemma 6, 5, and 7, Theorem 3 holds. $\square$

## D DATA GENERATION

In this section, we discuss the data generation process to form the dataset $\mathcal{D}$ of size $n_{\text{data}}$ consisting of input and output pairs. We take the input in the mapping $a(x) := u_0(x)$ to be the initial condition. We first discretize the initial condition $a(x)$ on a spatial grid of resolution $N$, i.e. $a_i := a(x_i)$ for $x_i \in \Omega, i = 0, \ldots N-1$. We then discretize the exact or reference solution $u(x,t)$ on a spatio-temporal grid of size $N \times N_t$, i.e. $u_{i,j} := u(x_i, t_j)$ for $x_i \in \Omega, i = 0, \ldots, N-1$, and $t_j < t_{j+1} \in \mathbb{R}_+, j = 1, \ldots, N_t$ with fixed spatial step sizes $\Delta x = (x_{N-1} - x_0)/N$, and temporal step size $\Delta t = t_{N_t}/N_t$ on a time interval $[0, t_{N_t}]$. We construct the dataset using the solution at the last $M$ time steps, that is, at times $t_{N_t-M+1} < \cdots < t_{N_t} \in \mathbb{R}_+$. We take different $N_t$ based on the dimensionality of the problem (See Table 7). We take $M = 1$ for single-step predictions, and $M = 25$ for multi-step predictions. In the following subsections, we discuss the selection over various input and output pairs, by uniformly sampling different PDE parameters in the initial condition.

| Dimension of Problem | $n_{\text{data}}$ | $n_{\text{train}}$ | $n_{\text{test}}$ | $N_t$ | $M$ |
|---|---|---|---|---|---|
| 1D space | 600 | 500 | 100 | 1 | 1 |
| 2D (1D space + time) | 1200 | 1000 | 200 | 200 | 25 |
| 3D (2D space + time) | 1200 | 1000 | 200 | 30 | 25 |

Table 7: Data generation sizes and parameters for different dimensions (1D, 2D and 3D). Multi-step predictions are included in the 2D and 3D problems.

## D.1 1D Stokes' second problem

The one-dimensional Stokes' second problem is given by the heat equation in the $y-$coordinate as:

$$
\begin{aligned}
u_t &= \nu u_{yy}, & y \in [0,1], t \geq 0, \\
u_0(y) &= U e^{-ky} \cos(ky), & y \in [0,1], \\
u(y=0,t) &= U \cos(\omega t), & t \geq 0,
\end{aligned}
\tag{49}
$$

with viscosity $\nu \geq 0$, oscillation frequency $\omega$, $k = \sqrt{\omega/(2\nu)}$, $U \geq 0$, and analytical solution:

$$
u_{\text{exact}}(x,t) = U e^{-ky} \cos(ky - \omega t). \tag{50}
$$

We construct the dataset $\mathcal{D}$ by fixing $\nu \in \{0.1, 0.02, 0.005, 0.002, 0.001\}, U = 2$ and sampling $\omega$ uniformly in $[3,4]$ to obtain the input and output pairs, consisting of the corresponding discretizations of the initial condition in eq. (49) and exact solution in eq. (50), respectively. The spatial resolution is $N = 500$, and temporal resolution is $N_t$ for $t \in [0,2]$.

## D.2 1D Burgers' Equation

The one-dimensional viscous Burgers' equation is given as:

$$
u_t + (u^2/2)_x = \nu u_{xx}, \qquad x \in [0,1], t \geq 0, \tag{51}
$$

with non-linear and convex flux function $f(u) = u^2/2$, and viscosity $\nu \geq 0$. The diffusive term $\nu u_{xx}$ on the right hand side of eq. (51) is added to smooth out the sharp transitions in $u(x,t)$. Depending on the initial condition, the solution can contain shocks and/or rarefaction waves that can be easily seen with classical Riemann problems (LeVeque (1992)). In various applications, it is critical that the numerical solution has the correct shock speed $dx/dt = f'(u) = u$. For example, consider an aircraft wing flying at a high enough speed for the flow to develop sonic shocks. The pressure distribution on the aircraft wing, which determines the lift and drag forces, is highly sensitive to the shock location.

### D.2.1 Dirichlet Boundary condition

We consider various Riemann problems for which the initial condition $u_0(x)$ is given as:

$$
u_0(x) = \begin{cases} u_L, & \text{if } x \leq 0.5, \\ u_R, & \text{if } x > 0.5, \end{cases} \tag{52}
$$

where $u_L, u_R \in \mathbb{R}$. The exact solution of eq. (51), with initial condition in eq. (52) is given as:

$$
u_{\text{exact}}(x,t) = \frac{u_L + u_R}{2} - \frac{u_L - u_R}{2} \tanh\left(\frac{(x - 0.5 - st)(u_L - u_R)}{4\nu}\right), \tag{53}
$$

where $s = (u_L + u_R)/2$ denotes the shock speed. We prescribe time-varying Dirichlet boundary conditions on both boundaries, i.e. $u(0,t) = u_{\text{exact}}(0,t)$ and $u(1,t) = u_{\text{exact}}(1,t)$, $\forall t \geq 0$. We choose $u_L > u_R$ leading to the formation of a shock in the solution $u(x,t)$ of eq. (51). We construct the dataset $\mathcal{D}$ by fixing $u_R = 0$, and varying the value of $u_L = w + \epsilon\mu$, where $w = 0.8$, $\epsilon = 0.01$ and $\mu \sim \mathcal{N}(0,1)$ to generate different initial conditions in eq. (52), and exact solutions in eq. (51) for $t \in [0, 1.2]$.

### D.2.2 Periodic boundary condition

We construct the dataset $\mathcal{D}$ using a finite difference reference solution of Burgers' equation for every random sample $u_0(x)$ and $t \in [0,1]$ as done in Li et al. (2020a).

## D.3 2D Navier-Stokes Equation

We consider a two-dimensional lid-driven cavity problem, which consists of a square cavity filled with fluid. The behavior of the fluid is described by the incompressible Navier-Stokes equations given as:

$$
\begin{aligned}
\vec{u}_t + \vec{u} \cdot \nabla\vec{u} &= -\nabla p + (1/Re)\nabla^2\vec{u}, & x, y \in [0,1]^2, t \geq 0, \\
\nabla \cdot \vec{u} &= 0, & x, y \in [0,1]^2, t \geq 0,
\end{aligned}
\tag{54}
$$

where $\vec{u} = \begin{bmatrix} u^{(1)}, & u^{(2)} \end{bmatrix}^T$ denotes the 2D velocity vector, $p$ the pressure and $\nu$ the viscosity. The initial condition is given as $u_0(x,y) = [U, \quad 0]^T \mathbb{1}(y = 1)$ for $x, y \in [0,1]^2$. We enforce Dirichlet conditions, i.e. $\vec{u}(x,1,t) = [U, \quad 0]^T$ for $U > 0$, $\vec{u}(x,0,t) = \vec{u}(0,x,t) = \vec{u}(1,y,t) = \vec{0}$. Figure 7 illustrates that at the top boundary, a tangential velocity is applied to drive the fluid flow in the cavity, and the remaining three walls are defined as no-slip boundary conditions with 0 velocity. For computational efficiency, we express the Navier-Stokes equation in terms of the scalar $z$ component of the vorticity $w = (\nabla \times \vec{u}) \cdot [0,0,1]^T$ in eq. (6) as done in Li et al. (2020a). We construct the dataset $\mathcal{D}$ by varying the value of $U$ in the initial condition uniformly in $[1, 1.5]$ to generate the different samples of the initial condition and the corresponding reference solution of eq. (54). We use the projection method (Chorin (1968)) on a staggered grid with $N^2$ grid points $(x_i, y_j)$ with $i, j = 1, ..., N$ to compute the reference solution. The pressure field is computed by solving a Poisson equation using a second order central discretization, to enforce the incompressibility condition. For the time-stepping schemes, we use a second-order Adams-Basforth method for the non-linear term and a Crank-Nicolson method for the diffusive term (LeVeque (1992)) for $N_t = 30$ timesteps over the interval $[0, 2]$. Lastly, we compute $w = \partial u^{(2)}/\partial x - \partial u^{(1)}/\partial y$ using finite differences.

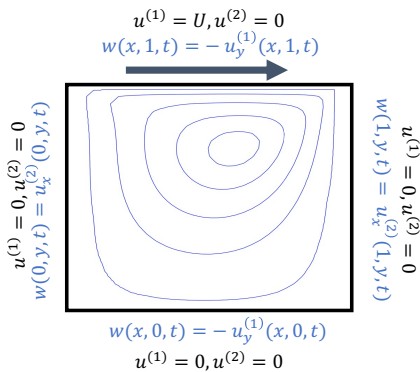

Figure 7: No-slip boundary conditions for the lid-driven cavity problem and streamlines showing the direction of the fluid flow.

### D.4 1D HEAT EQUATION

The one-dimensional heat equation is given as:

$$
\begin{aligned}
u_t - k u_{xx} &= f(x,t), && x \in [0,1], t \geq 0, \\
u_0(x) &= \cos(\omega \pi x), && x \in [0,1],
\end{aligned}
\tag{55}
$$

where $k > 0$ denotes the conductivity, $f(x,t)$ denotes a heat source, and $\omega \in \mathbb{R}$. We prescribe Neumann boundary conditions, i.e. $u_x(0,t) = 0$ and $u_x(1,t) = U \sin \pi t$ with $U \in \mathbb{R}$, and choose $f(x,t) = U\pi \frac{x^2}{2} \cos \pi t$. The exact solution of eq. (55) is given as:

$$
u_{\text{exact}}(x,t) = U \frac{x^2}{2} \sin(\pi t) - \frac{Uk}{\pi}(\cos(\pi t) - 1) + \frac{\sin(\omega \pi)}{\omega \pi} + \sum_{n=1}^{\infty} a_n \cos(n\pi x) e^{-k(n\pi)^2 t}, \tag{56}
$$

where

$$
a_n = \begin{cases} \sin[(\omega + n)\pi]/[(\omega + n)\pi] + \sin[(\omega - n)\pi]/[(\omega - n)\pi], & \text{if } \omega \neq n, \\ 1, & \text{if } \omega = n, \end{cases} \tag{57}
$$

We construct the dataset $\mathcal{D}$ by sampling $\omega$ uniformly in $[2.01, 3.99]$ with fixed $k = 0.01$ and $U = 5$ to obtain the input and output pairs, consisting of the corresponding discretizations of the initial condition and exact solution in eq. (56), respectively on a grid with resolution $N$. The temporal resolution is given by $N_t$ on the time interval $[0, 2]$.

### D.5 2D WAVE EQUATION

The two-dimensional wave equation is given as:

$$
u_{tt} = c^2(u_{xx} + u_{yy}), \qquad x, y \in [0,1]^2, t \geq 0, \tag{58}
$$

where $c \in \mathbb{R}_{\geq 0}$ denotes the wavespeed. We impose the following Neumann boundary conditions: $u_x(0,y,t) = u_x(1,y,t) = u_y(x,0,t) = u_y(x,1,t) = 0$, for $t \geq 0$. The initial conditions are given as: $u_0(x,y) = k \cos(\pi x) \cos(\pi y)$, with $k \in \mathbb{R}$ and $u_t(x,y,0) = 0$. The exact solution of eq. (58) is given as:

$$
u_{\text{exact}}(x,t) = k \cos(\pi x) \cos(\pi y) \cos(c\sqrt{2}\pi t). \tag{59}
$$

We construct the dataset $\mathcal{D}$ by fixing $c = 1$, and sampling $k$ uniformly in $[3, 4]$ to obtain the input and output pairs, consisting of the corresponding discretizations of the initial condition and exact solution in eq. (59), respectively on a two-dimensional uniform mesh with resolution $N^2$, i.e. $(x_i, y_j)$ with $i, j = 1, ..., N$. The temporal resolution is given by $N_t$ on the time interval $[0, 2]$.

## E  EXPERIMENTAL SETUP

In this section, we list the detailed experiment setups and parameter searches for each experiment. We incorporate time into the problem dimension for multi-step predictions. Table 8 shows the parameters values selected for BOON and FNO. For PINO, we conduct a multi-dimensional grid search, over the values $\{0, 0.0001, 0.001, 0.01, 0.1, 1\}$, to find the optimal weights attributed to the PDE, boundary and initial condition losses and report the minimum $L^2$ errors in all our tables. We use the default hyper-parameters for the other benchmarks in our experiments.

| Dimension of Problem | Modes | Channels |
|---|---|---|
| 1D space | 16 | 64 |
| 2D (1D space + time) | 12 | 32 |
| 3D (2D space + time) | 8 | 20 |

Table 8: Training parameters in BOON for problems of different dimensions (1D, 2D and 3D).

## F  ADDITIONAL EXPERIMENTS

In this section, we include additional experimental results, which show that our proposed BOON has the lowest relative $L^2$ error compared to the state-of-the-art baselines, and has the lowest absolute $L^2$ boundary errors on several single-step ($M = 1$) and multi-step ($M = 25$) predictions.

### F.1  DIRICHLET BOUNDARY CONDITION

#### F.1.1  1D STOKES' SECOND PROBLEM

We are interested in learning the operator $T : u_0(y) \to u(y, t)$, $t = 2$, where $u_0(y) = Ue^{-ky} \cos(ky)$ and $k = \sqrt{\omega/2\nu}$ for the one-dimensional Stokes' second problem with Dirichlet boundary conditions (See Appendix D.1). Table 9 shows the results for single-step prediction (See Table 2 for the multi-step prediction results).

| Model | $\nu = 0.1$ | $\nu = 0.02$ | $\nu = 0.005$ | $\nu = 0.002$ | $\nu = 0.001$ |
|---|---|---|---|---|---|
| BOON (Ours) | **0.0189 (0)** | **0.0370 (0)** | **0.0339 (0)** | **0.0354 (0)** | **0.0370 (0)** |
| FNO | 0.0199 (0.0043) | 0.0410(0.0093) | 0.0578 (0.0135) | 0.0370 (0.0077) | 0.0557 (0.0116) |
| APINO | 0.1343 (0.0914) | 0.1398 (0.0636) | 0.0972 (0.0271) | 0.0898 (0.0181) | 0.0986 (0.0172) |
| MGNO | 0.28457 (0.03530) | 0.25150 (0.03778) | 0.25007 (0.03574) | 0.1659 (0.0208) | 0.1609 (0.0238) |

Table 9: **Single-step prediction for Stokes' with Dirichlet BC**. Relative $L^2$ error ($L^2$ boundary error) for Stokes' second problem with varying viscosities $\nu$ at resolution $N = 500$ and $M = 1$.

#### F.1.2  1D BURGERS' EQUATION

We are interested in learning the operator $T : u_0(x) \to u(x, t)$, $t \in [0, 1.2]$, where $u_0(x)$ is defined in eq. (52) for the one-dimensional viscous Burgers' equation with Dirichlet boundary conditions (See Appendix D.2.1). Table 10 shows the multi-step prediction results, where we train our model to predict solutions at the last $M = 25$ time steps (See Table 3 for the single-step prediction results).

| Model | $\nu = 0.1$ | $\nu = 0.05$ | $\nu = 0.02$ | $\nu = 0.005$ | $\nu = 0.002$ |
|---|---|---|---|---|---|
| BOON (Ours) | **0.00040 (0)** | **0.00039 (0)** | **0.00036 (0)** | **0.00023 (0)** | **0.00045 (0)** |
| FNO | 0.00043 (0.00017) | 0.00050 (0.00027) | 0.00060 (0.00041) | 0.00088 (0.00061) | 0.00120 (0.00039) |
| PINO | 0.01232 (0.05581) | 0.01318 (0.06491) | 0.01583 (0.08101) | 0.02243 (0.11506) | 0.02358 (0.12070) |
| DeepONet | 0.00577 (0.00139) | 0.00459 (0.00120) | 0.00540 (0.00178) | 0.00633 (0.00150) | 0.00723 (0.00161) |

Table 10: **Multi-step prediction for Burgers' equation with Dirichlet BC**. Relative $L^2$ error ($L^2$ boundary error) for various benchmarks with varying viscosities $\nu$, and $M = 25$.

## F.2 PERIODIC BOUNDARY CONDITION

### F.2.1 1D BURGERS' EQUATION

We are interested in learning the operator $T : u_0(x) \rightarrow u(x, t)$, where $u_0 \sim \mathcal{N}(0, 625(-\Delta + 25\ I)^{-2})$, and $\Delta$ and $I$ denote the Laplacian and identity matrices, respectively, for the one-dimensional viscous Burger's equation with periodic boundary conditions, as done in Li et al. (2020a) (See Appendix D.2.2). Table 11 shows the results for the single-step prediction, where $t = 1$. Table 12 shows the multi-step predictions, where $t \geq 0$, $M = 25$ and final time $t_M = 1$. We see that BOON has the highest performance compared to the other baselines across various resolutions $N$. As expected with Neural Operators, we see that the BOON error is approximately constant, as we increase the grid resolution $N$. In addition, BOON achieves a small error in the low resolution case of $N = 32$, showing its effectiveness in learning the function mapping through low-resolution data.

| Model | $N = 32$ | $N = 64$ | $N = 128$ | $N = 256$ | $N = 512$ |
|---|---|---|---|---|---|
| BOON (Ours) | **0.00646 (0)** | **0.00243 (0)** | **0.00221 (0)** | 0.00176 (**0**) | 0.00132 (**0**) |
| FNO | 0.0066 (0.00046) | 0.0028 (0.00078) | 0.0029 (0.00161) | 0.0021 (0.00147) | 0.0014 (0.00081) |
| APINO | 0.0121 (0.00822) | 0.0045 (0.00452) | 0.0023 (0.00236) | **0.0016** (0.00143) | **0.0013** (0.00075) |
| MGNO | 0.0390 (0.0137) | 0.0473 (0.00751) | 0.0606 (0.00786) | 0.0706 (0.00702) | 0.0807 (0.01109) |

Table 11: **Single-step prediction for Burgers' equation with Periodic BC**. Relative $L^2$ error ($L^2$ boundary error) for various benchmarks with varying resolutions $N$, viscosity $\nu = 0.1$, and $M = 1$.

| Model | $N = 32$ | $N = 64$ | $N = 128$ |
|---|---|---|---|
| BOON (Ours) | **0.00097 (0)** | **0.00093 (0)** | **0.00101 (0)** |
| FNO | 0.00133 (0.00160) | 0.00104 (0.00175) | 0.00201 (0.01152) |
| PINO | 0.00170 (0.00087) | 0.00153 (0.00020) | 0.00175 (0.00005) |
| DeepONet | 0.08857 (0.00356) | 0.08805 (0.00529) | 0.08615 (0.00733) |

Table 12: **Multi-step prediction for Burgers' equation with Periodic BC**. Relative $L^2$ error ($L^2$ boundary error) for various benchmarks with varying resolutions $N$, viscosity $\nu = 0.1$ and $M = 25$.

## F.3 NEUMANN BOUNDARY CONDITION

### F.3.1 1D HEAT EQUATION

We are interested in learning the operator $T : u_0(x) \rightarrow u(x, t)$, $t = 2$, where $u_0(x) = \cos(\omega \pi x)$ for the one-dimensional heat equation with Neumann boundary conditions (See Appendix D.4). Table 13 shows the results for the single-step prediction (See Table 5 for the multi-step prediction results).

| Model | $N = 50$ | $N = 100$ | $N = 250$ | $N = 500$ |
|---|---|---|---|---|
| BOON (Ours) | **0.00901 (0)** | **0.01070 (0)** | **0.01170 (0)** | **0.00811 (0)** |
| FNO | 0.01636 (0.43876) | 0.01711 (0.48925) | 0.01694 (0.50789 ) | 0.01730 (0.50619 ) |
| APINO | 0.10457 (0.24939) | 0.09128 (0.33732) | 0.07999 (0.21998) | 0.08310 (0.21914) |
| MGNO | 0.04685 (0.15536) | 0.05410 (0.5251) | 0.07314 (1.7357) | 0.05796 (1.3783) |

Table 13: **Single-step prediction for the heat equation with Neumann BC**. Relative $L^2$ error ($L^2$ boundary error) for various benchmarks with varying resolutions $N$ and $M = 1$.

### F.4 PREDICTION AT HIGH RESOLUTION FOR HEAT EQUATION

In this section, we evaluate the resolution extension property of BOON using the dataset generated from heat equation with Neumann boundary conditions dataset (See Appendix D.4). We train BOON on a dataset $\mathcal{D}_{\text{train}}$ with data of resolution $N \in \{256, 512, 1024\}$. We test our model using a dataset $\mathcal{D}_{\text{test}}$ and finite difference coefficients $c_j$ generated using a resolution $N \in \{1024, 2048, 4096\}$. Table 14 shows that on training with a lower resolution, for example, $N = 1024$, the prediction error at $4X$ higher resolution $N = 4096$ is 0.01251 or 1.25%. Also, learning at an even coarser resolution of $N = 512$, the proposed model can predict the output of data with $8X$ times the resolution, i.e. $N = 4096$, with a relative $l_2$ error of 2.16%.

| Train \ Test | $N = 1024$ | $N = 2048$ | $N = 4096$ |
|---|---|---|---|
| $N = 256$ | 0.07197 | 0.08051 | 0.08447 |
| $N = 512$ | 0.01527 | 0.01947 | 0.02159 |
| $N = 1024$ | 0.00813 | 0.01086 | 0.01251 |

Table 14: **Resolution invariance for Neumann BC with single-step prediction**. BOON trained at lower resolutions can predict the output at higher resolutions.

### F.5 KERNEL CORRECTION APPLIED TO MULTI-WAVELET BASED NEURAL OPERATOR

In this section, we show our proposed approach of kernel correction can also be applied to other kernel-based neural operators, e.g. the Multiwavelet-based neural operator (MWT) (Gupta et al. (2021)). We evaluate the performance of MWT and MWT-BOON on Stokes' second problem and Burgers' equation with Dirichlet boundary conditions (See Appendices D.1, D.2.1, respectively). Table 15 16 show that BOON operations also improve the MWT-based neural operator.

| Model | $\nu = 0.1$ | $\nu = 0.05$ | $\nu = 0.02$ | $\nu = 0.005$ | $\nu = 0.002$ |
|---|---|---|---|---|---|
| BOON-FNO | **0.00012 (0)** | **0.00010 (0)** | **0.000084 (0)** | **0.00010 (0)** | 0.00127 (0) |
| BOON-MWT | 0.00020 (0) | 0.00025 (0) | 0.00022 (0) | 0.00020 (0) | **0.00034 (0)** |
| FNO | 0.0037 (0.0004) | 0.0034 (0.0004) | 0.0028 (0.0005) | 0.0043 (0.0004) | 0.0050 (0.0022) |
| MWT | 0.00020 (5.25e-6) | 0.00020 (2.44e-05) | 0.00026 (8.93e-05) | 0.00027 (0.00030) | 0.00054 (0.00013) |
| APINO | 0.0039 (0.0004) | 0.0040 (0.0005) | 0.0037 (0.0007) | 0.0048 (0.0008) | 0.006 (0.0016) |
| MGNO | 0.0045 (0.0004) | 0.0046 (0.0005) | 0.0048 (0.0008) | 0.0064 (0.0015) | 0.0101 (0.0006) |
| DeepONet | 0.00587 (0.00124) | 0.0046 (0.00130) | 0.00580 (0.00128) | 0.00623 (0.00140) | 0.00773 (0.00141) |

Table 15: **Single-step prediction for Burgers' with Dirichlet BC**. Relative $L^2$ test error ($L^2$ boundary error) for Burgers' equation with varying viscosities $\nu$ at resolution $N = 500$ and $M = 1$.

| Model | $\nu = 0.1$ | $\nu = 0.02$ | $\nu = 0.005$ | $\nu = 0.002$ | $\nu = 0.001$ |
|---|---|---|---|---|---|
| BOON-FNO | **0.0189 (0)** | **0.0370 (0)** | **0.0339 (0)** | **0.0354 (0)** | **0.0370 (0)** |
| BOON-MWT | 0.0701 (0) | 0.0602 (0) | 0.0631 (0) | 0.0579 (0) | 0.07992 (0) |
| FNO | 0.0199 (0.0043) | 0.0410 (0.0093) | 0.0578 (0.0135) | 0.0370 (0.0077) | 0.0557 (0.0116) |
| MWT | 0.0796 (0.0125) | 0.1080 (0.0150) | 0.0940 (0.0164) | 0.0889 (0.0130) | 0.1060 (0.0176) |
| APINO | 0.1343 (0.0914) | 0.1398 (0.0636) | 0.0972 (0.0271) | 0.0898 (0.0181) | 0.0986 (0.0172) |
| MGNO | 0.28457 (0.03530) | 0.25150 (0.03778) | 0.25007 (0.03574) | 0.1659 (0.0208) | 0.1609 (0.0238) |

Table 16: **Single-step prediction for Stokes' with Dirichlet BC**. Relative $L^2$ error ($L^2$ boundary error) for Stokes' second problem with varying viscosities $\nu$ at resolution $N = 500$ and $M = 1$.

