# OpenReview forum: "Guiding continuous operator learning through Physics-based boundary constraints"
_ICLR.cc/2023/Conference — ICLR 2023 poster_

### Official Review · Reviewer_2TdD · 2022-10-21

**Confidence:** 3
**Correctness:** 3
**Technical Novelty And Significance:** 3
**Empirical Novelty And Significance:** 3
**Recommendation:** 6

**Clarity, Quality, Novelty And Reproducibility:**


Clarity moderate. Quality good. Novelty good. Reproducibility good.


**Strength And Weaknesses:**


s: Principled method.

s: The results are convincing. The method achieves perfect behavior at boundaries (as expected), and also shows moderate interim accuracy improvement.

w: This paper is incremental since it does an “FNO + BC” type model. The paper doesn’t motivate or demonstrate why adding BCs is important. I’m not sure if this is that important feature, and the author’s should more clearly argue why it’s important to include boundary conditions in FNOs.

w: There are clarity issues. The algorithms 1-3 are the key contribution of this paper, and yet they are gibberish to me with no explanations. I can’t really follow the main idea of how the kernels are corrected.


Technical comments
* Since PDEs are not linear in reality, how come it makes sense to still use the kernel form eq 1? I don’t think the “note” explains this
* What does L^2 mean in eq 2? a(y) lives in \A space, so how do we define a product between L^2 and \A instances?
* What does s and p denote?
* In eq 1 G is a function, in eq 3 it's an operator. Is this intentional?
* G only applies to values at boundary, by definition. In eq 3 we apply G to solutions of any points a \in A. The G is thus mostly undefined in eq 3.
* The U_bdy is an intersection of space and its boundary. Isn’t that then trivially just the boundary? Is this a typo?
* What about the boundary of A, why is that not considered (around eq 3 and fig2)
* How come in \D we only have pairs from A and U_bdy. Why can’t we have points from boundary here? Is it intentional that this now restricts A implicitly (or does it)?
* || is L2 error at boundary, yet most points inside the sum in eq 3 are not at the boundary. I don’t understand. Do you then skip some of the points inside the sum?
* If we have boundary condition \alpha=0, doesn’t that lead to division-by-zero in prop1?
* I would expect a common boundary condition is that we have a time-invariant fixed value at boundary. This would then lead to a Dirichlet kernel that is simply an indicator function. Surely this is not a useful kernel and does nothing. What would we do then?


**Summary Of The Paper:**

This paper adds support for boundary conditions to FNO’s, which are neural operators that learn the solution map of unknown PDE systems.


**Summary Of The Review:**

This is a solid paper that succesfully includes proper boundary behavior to FNO models.

---

> ### Author Response · Authors · 2022-11-11
> **Thank you! And clarifications (1/2)**
>
> We would like to thank the reviewer for giving a detailed read to our paper. We also express our gratitude for acknowledging the novelty and calling our work a “solid paper”! We now take the opportunity to point-wise address the clarifications that the reviewer has asked for.
>
> **Applicability of our approach to any kernel-based Neural Operator** We would like to emphasize that the proposed work serves two purposes: (i) explore the novel problem of incorporating boundary values to make neural operators (NO) \textit{physically relevant/useful} as well as orders-magnitude more accurate (2X-20X improvement), and (ii) propose an efficient and principled solution for the guaranteed satisfaction of the boundary constraints. We show that Theorem-1, 2, and Algorithms-1,2,3 deliver the strength of a given base NO, like cost complexity, while giving order-high accuracy and physical relevance of the solution. Next, our results in Theorem-1 are general for any kernel-based NO, for example, FNO (Li et al. (2020a)), and MWT (Gupta et al. (2021)). We rigorously evaluate the results in an exhaustive setting of all three boundary conditions with various settings of dimensions using FNO for the consistency of results. Additionally, we have already shown in Appendix E.5 that our approach seamlessly integrates with other NO like MWT, where the findings are consistent with Section-4.
>
> Finally, we would like to bring the reviewer’s attention to Figure-1 where we have demonstrated the importance of incorporating the boundary conditions. In the Figure-1 caption, we explain that violating such constraints makes a model (like FNO) obtain a solution suggesting that heat is flowing across an insulator, rendering the solution physically irrelevant. In contrast, we show that our proposed approach (BOON) obeys the physical constraints of Neumann boundary conditions at both boundaries.
>
> **Explanations for the algorithms** We thank the reviewer for raising the clarification concern. This has led us to further add clarifications that explain the algorithms better. We have expanded the arithmetic operations of Theorem-1 in Appendix-C.2 and shown how the algorithms are a simple outcome of our main Theorem-1.
>
> **Technical comments clarification** We point-wise address the technical queries below.
>
> (1) This work is concerned with making kernel corrections for eq. (2). We agree with the reviewer that PDEs operators are non-linear in general and we have already mentioned this in Section-2.2 note. We would like to emphasize that working with a single kernel as in eq. (2) and then adding multiple layers with non-linearity (e.g., GeLU) is the standard setting as in all of the existing kernel-based NO, for example, FNO (Li et al. (2020a)), MWT (Gupta et al. (2021)). This setting is not new to our BOON work but is taken by all these existing works.
>
> (2)  We choose $\mathcal{A}$ to be a Sobolev space $\mathcal{H}^{s,2}$ which is a subspace of the $L^2$ Banach function space. So, the kernel $K$ is well-defined. We further added a sentence in the paper to indicate that we choose $p = 2$ and also provided an elaborated list of notations in Appendix-A page-14 for enhanced readability.
>
> (3) A Sobolev space $\mathcal{H}^{s,p}$ is defined as the subset of functions $h \in L^p(\Omega)$ such that $h$ and its weak derivatives, up to order $s > 0$, have a finite $L^p$ norm, $(p \geq 1)$ (see Ref [R3.1]). We added Appendix-A with a summary of the notations on page 14.
>
> (4) We would like to point out that we have not mentioned anywhere in the current work that $\mathcal{G}$ is a function. After its introduction in eq. (1), we define $\mathcal{G}$ as an operator in the text following eq. (1) on page 3. From the author’s understanding, the confusion may have arisen from the extra brackets in the eq. (1) after $\mathcal{G}$. While this notation is also correct, for better understanding, we have omitted the brackets in eq. (1).
>
> (5) $\mathcal{G}$ is an operator that applies to a function across the entire domain, however, it impacts only the boundary points $\partial\Omega$, therefore, $\mathcal{G}u$ is well-defined. Note that, in eq. (3) we are evaluating a boundary norm, so only boundary points will be included in the integral computation as already defined by $(\int\nolimits_{\partial\Omega}\vert\vert . \vert\vert^2 dx)^{1/2}$, or integral over $\partial \Omega$.
>
> (6) $\mathcal{U}_{\text{bdy}}$ is an intersection of $\mathcal{U}$ which is a function space of PDE solutions and $\partial \mathcal{U}$ which is a space of boundary-satisfied functions (but not necessarily the PDE solutions). Therefore, their intersection $\mathcal{U}bdy$ denotes a space of functions that are PDE solutions as well as satisfy boundary constraints. We further elaborate this by modifying Figure-2 a little as well as adding elaborated notations in Appendix-A page-14.

---

> > ### Author Response · Authors · 2022-11-11
> > **Thank you! And clarifications (2/2)**
> >
> > (7) $\mathcal{A}$ is the input function space. Since the input is always known, and we require that boundary conditions have to be satisfied for all time including $t=0$ (eq. (1)), therefore, we omit defining the boundary for $\mathcal{A}$ because it is satisfied implicitly.
> >
> > (8) Note that the boundary points are implicit in the definition of $\mathcal{U}_{\text{bdy}}$ as it is an intersection of $\mathcal{U}$ and **$\partial\mathcal{U}$**.
> >
> > (9) When computing the boundary $L^2$ error or $\vert\vert . \vert\vert_{\partial\mathcal{H}^{s,p}}$, the domain of integration is $\partial\Omega$ as described in the definition $(\int\nolimits_{\partial\Omega}\vert\vert . \vert\vert^2 dx)^{1/2}$. Therefore, all interior points will be ignored which is a requirement for computing boundary error.
> >
> > (10) The reviewer has raised an important point of dealing with zero Dirichlet boundary conditions for the input. Computationally, an explicit division is never required which is what we have already implemented in our uploaded code with the paper. We have taken this opportunity to also clarify this explicitly in Algorithm-1,2 and also added some explanation in Appendix-C.2. We have also added a sentence in the paragraph following Proposition-1 on page-4, and its explanation in the Proposition-1 proof in Appendix-B page-14 as well. In addition, we added a sentence that a similar argument also holds for the special case where $u_0(x_0)=0$ for Neumann on page 15. We would like to thank the reviewer for bringing our attention to this!
> >
> > (11) The proposed kernel correction algorithm does not modify the kernel on the **interior rows**. In the case that a fixed Dirichlet boundary condition is prescribed at $x_0$, only the first row of $K$ is an indicator function, i.e, $K(x_0, y) =  1(y=x_0)$ (See Lemma 1 in Appendix C.1.1). In our experiments with Dirichlet and Burgers’ equation (Section 4.1.2), the right boundary condition can be written as fixed 0 Dirichlet for all times when the shock is in the domain. This is a common ``no-slip” boundary condition in practice.
> >
> > References:
> >
> > [R3.1] L.C. Evans. Partial Differential Equations. Graduate studies in mathematics. American Mathematical 335 Society, 2010. ISBN 9780821849743. URL https://books.google.com/books?id= 336 Xnu0o_EJrCQC.

---

> > ### Comment · Reviewer_2TdD · 2022-11-11
> > **motivation**
> >
> > In my review I raised the concern of motivation: why is adding BCs to FNOs significant?
> >
> > I fully agree that boundary conditions are important for a physics person modelling physics systems, such as conductors or insulators. However, ICLR is a machine learning conference, and we are talking about neural PDE systems, which are generally intended to model complex or messy systems where principled mechanical modelling can't cope. Thus, FNOs are not going to be used for insulators and conductors, but instead their applications lie in climate modelling, images and video (eg. DALLE), brain dynamics, chemical system simulation, etc.
> >
> > Can you explain why BCs are important in these kind of applications? My limited understanding of BCs is that they are basically the "walls" around the system. However, in eg. brain data these walls do not even appear. One application might be natural images, where clearly we don't want the information to flow outside the image (eg. in recent PDE-inspired generative models, eg. Hoogeboom'22).

---

> > > ### Author Response · Authors · 2022-11-12
> > > **Motivation follow-up (1/3)**
> > >
> > > We emphasize that our proposed boundary constrained neural operators are a new class of NOs which satisfy constraints while obtaining a solution. Constrained problems are definitely more challenging than their unconstrained counterparts but at the same time guarantee uniqueness as well as meaningful physical interpretation. The example of an insulator in Figure 1 is a simple attempt to illustrate the physical importance of incorporating boundary conditions (BCs).  In the introduction, we also provide additional examples of no-slip boundary conditions for wall-bounded viscous flows, periodic boundary conditions for modeling isotropic homogeneous turbulent flows.  Additionally, we have taken much more complicated BCs, for example, time-varying lid-cavity BC which drives the flow in Section 4.1.3, moving boundaries- no-slip BC for viscous flows in Section 4.1.1, time-varying BC for viscous fluids in Section 4.1.2, and periodic BC for isotropic homogeneous turbulent flows in Section 4.3.
> > >
> > > First, we agree with the reviewer that modeling systems is challenging. However, “modeling a system” refers to discovering the underlying dynamics of the system ($\mathcal{F}$ in eq. (1)). And this is, in fact, the strength of our work, because being a data-driven approach we never require the explicit knowledge of the underlying physical dynamics. Whereas, boundary conditions are seamlessly available pieces of information that any model should satisfy in order to predict the underlying dynamics of the system accurately. Regarding the problem complexity, the Navier-Stokes (NS) equations are the most complicated PDEs in the field of fluid dynamics and emerge in any problem that involve fluid flow such as *climate forecasting*, turbulent flow modeling for aerodynamic designs, combustion engines, heat transfer, energy transport modeling in batteries, cardiovascular flows, etc. We chose the lid-driven cavity flow with time-varying Dirichlet BC in Section 4.1.3 as one of the commonly used examples of NS equations to model flow recirculation phenomenon in confined domains. This problem setting has allowed study of vortical hydrodynamic patterns which occur in numerous real world phenomena such as ocean flow circulations, fluid mixing systems for drug delivery and food processing, and flow circulations in the wake behind a car.
> > >
> > > Second, for any problem where the domain $\Omega$ of the dynamics is finite, the boundaries $\partial\Omega$ are natural to appear. The boundaries are not “walls” in the sense that they only limit the domain of the current underlying dynamics $\mathcal{F}$, and most of the time they serve as an interaction medium with other neighboring domains. Therefore, these values can be space and time-varying, as we have seen in Section 4. From the machine learning perspective, gathering boundary information can be much easier to obtain than data on the complete domain. For example, in hydrology [R3.2], monitoring soil moisture dynamics vs surface depth around a river requires placing probes inside the soil. While it is much easier to monitor the surface moisture (boundary value) rather than the moisture at depths due to obvious sensor failures and maintenance issues. In such cases, boundary value data can ease-up the data-driven based operator learning process, and make the solution more accurate as well as data-efficient. It is out-of-scope of the current work to discuss and study all such cases, but it is an interesting future work that has tremendous practical applications!
> > >
> > > Third, we also provide references to recent Neural Operator papers in ML conferences, where they consider a subset of the exact PDEs that we provide in our paper, showing that these physical problems do have important applications within the ML community. Both Burgers’ equation with periodic BC and the Navier Stokes equation, which we tackled in our paper, were also considered by Li et al. [R3.6], published in ICLR 2021, Gupta et al. [R3.9], published in NeurIPS 2021 and Li et al., [R3.10]. Moreover, other physics-based PDEs, such as the Korteweg-de Vries (KdV) equation, Darcy Flow equation, advection equation, reaction-diffusion equation and Kuramoto-Sivashinsky (KS) equation also appeared in NeurIPS 2021, ICLR 2021 and ICLR 2022 [R3.6, R3.7, R3.8, R3.9, R3.10].

---

> > > > ### Author Response · Authors · 2022-11-12
> > > > **Motivation follow-up (2/3)**
> > > >
> > > > Fourth, an exact treatment of boundary conditions is necessary in a variety of fields including the ones that the reviewer has mentioned, such as brain dynamics, climate modeling and others as we discuss below:
> > > >
> > > > (a) A careful treatment of Neumann boundary conditions, which we consider in our work in Section 4.2, is required in various healthcare applications.
> > > >
> > > > (i) The growth of cancer cells in the brain, given by the diffusion-reaction equation, can be modeled using Neumann BCs. [R3.3]. The tumor growth and invasion of cancer cells into the surrounding tissue can be described as a boundary value problem which strongly depends on the actual values assigned on the physical boundary of cranium. Note that, in this case, the boundary is not modeling a “wall” separating the brain from the cranium but rather an intersection modeling interactions between them. This interaction is represented by the Neumann BC, and it is crucial to satisfy this BC correctly especially for very diffusive and invasive cancers like glioblastoma. A failure in accurately capturing the Neumann BC could yield incorrect conclusions about the rate of spreading of cancer in a human’s body leading to serious health consequences.
> > > >
> > > > (ii) BCs are also crucial for cardiovascular blood-flow modeling. A careful consideration of the outflow BCs, given by Neumann, is necessary to study unsteady back-flow. An unsteady backflow of blood, near the outlets (boundaries) of large vessels in the body, can lead to arrhythmias which can be serious or even life threatening. The authors [R3.4] also show how inconsistencies in the treatment of the Neumann BCs can lead to incorrectly modeling backflow.
> > > >
> > > > (b) We’ve also included references of boundary value problems in climate modeling (See Refs [R3.11], [R3.12], [R3.13]).  Note that boundary conditions are required in climate models for the values at the sea surface, including sea surface temperature (SST) and sea surface salinity (SSS). For example, future regional climate projections are impacted by boundary conditions. In West Africa, it was shown that small changes in the atmospheric humidity alone at the domain boundaries lead to a wetter Sahel due to the northward migration of rain belts during summer [R3.12]. The authors in the paper also show that the boundary conditions are essentially the sole factors that are leading to changes in atmospheric temperature. An exact satisfaction of the boundary conditions is critical for an accurate prediction of the climate especially when strong changes in the temperature are on the horizon and safety measures need to be taken.
> > > >
> > > > **Differences in modeling physics processes/underlying dynamics and image/video prediction**  Thank you for the recent reference to Hoogeboom ‘22 (https://arxiv.org/pdf/2209.05557.pdf). Yes, these diffusion based models have been very effective in denoising images and in image/video prediction. The key idea is to add noise to an image and use the heat equation in Eqn. 8 to smooth or blur the image. This is the same heat equation that we illustrated in Figure 1, and in the experiments in 4.2.1. It is one particular linear version of the general non-linear PDE $\mathcal{F}$ in eq.(1), which equals $\frac{d}{dt} - \Delta$ in this case. The heat/diffusion equation ($u_t - \Delta u = 0$) with applications in heat transfer and diffusion processes is a parabolic PDE and results in smoothed solutions. The reviewer may find it interesting that leveraging similar diffusion-like PDEs (level set equation under curvature flow) for images, MRI, and angiograms links back to Sethian et al. in 1995 [R3.5]. Similarly, adding diffusion is also commonly used in numerical methods to smooth artificial oscillations.  In our work, we are proposing a method that not only predicts smooth diffusive processes such as what we presented in 4.2.1, but also enables capturing complex non-linear physics such as turbulent flows in ocean dynamics as detailed with the Navier-Stokes equations above in Section 4.1.3. One key difference between image/video prediction and our work is that our model aims to learn an operator that enforces physical laws described by $\mathcal{F}$ and the boundary conditions. Whereas the patterns in an arbitrary image or a video may not carry any physical intuition. Additionally, the work in [R3.14] does consider a mechanical MNIST image dataset, which is used to study material deformation, and uses Dirichlet boundary conditions. This work also illustrates climate modeling using NO by mapping pressure and temperature.

---

> > > > > ### Author Response · Authors · 2022-11-12
> > > > > **Motivation follow-up (3/3)**
> > > > >
> > > > > References:
> > > > >
> > > > > [R3.2] Coon et al. Coupling surface flow and subsurface flow in complex soil structures using mimetic finite differences, Advances in Water Resources, Volume 144, 2020.
> > > > >
> > > > > [R3.3] Stamatakos GS, Giatili SG. A Numerical Handling of the Boundary Conditions Imposed by the Skull on an Inhomogeneous Diffusion-Reaction Model of Glioblastoma Invasion Into the Brain: Clinical Validation Aspects. Cancer Inform. 2017 Feb 3;16:1176935116684824. doi: 10.1177/1176935116684824. PMID: 28469383; PMCID: PMC5392020.
> > > > >
> > > > > [R3.4] Esmaily Moghadam, M., Bazilevs, Y., Hsia, TY. et al. A comparison of outlet boundary treatments for prevention of backflow divergence with relevance to blood flow simulations. Comput Mech 48, 277–291 (2011). https://doi.org/10.1007/s00466-011-0599-0
> > > > >
> > > > > [R3.5] Malladi, R., Sethain, J.A. Imaging processing via level set curvature flow, PNAS 92, pp. 7046-7050 (1995). (https://www.pnas.org/doi/epdf/10.1073/pnas.92.15.7046)
> > > > >
> > > > > [R3.6] Li et al., Fourier Neural Operator for Parametric Partial Differential Equations, ICLR, 2021.
> > > > >
> > > > > [R3.7] Gupta et al., Non-linear operator approximations for initial value problems, ICLR, 2022.
> > > > >
> > > > > [R3.8] Krishnapriyan et al., Characterizing possible failure modes in physics-informed neural works, Advances in Neural Information Processing Systems 34 (NeurIPS 2021).
> > > > >
> > > > > [R3.9] Gupta et al., Multiwavelet-based Operator Learning for Differential Equations, Advances in Neural Information Processing Systems 34 (NeurIPS 2021).
> > > > >
> > > > > [R3.10] Li et al., Physics-Informed Neural Operator for Learning Partial Differential Equations, 2021.
> > > > >
> > > > > [R3.11] Goergen, K. and Kollet, S., Boundary condition and oceanic impacts on the atmospheric water balance in limited area climate model ensembles, Scientific Reports, 11, 2021. (https://www.nature.com/articles/s41598-021-85744-y)
> > > > >
> > > > > [R3.12] Kim, J.H., Kim, Y., Wang, G., Impacts of boundary conditions on regional climate projections over West Africa, JGR Atmospheres, 2017.  https://doi.org/10.1002/2016JD026167
> > > > >
> > > > > [R3.13] Huber, M.B., Zanna, L., Drivers of uncertainty in simulated ocean circulation and heat uptake, Geophysical Research Letters, 2017. (https://agupubs.onlinelibrary.wiley.com/doi/10.1002/2016GL071587)
> > > > >
> > > > > [R3.14] Kissas et al., Learning Operators with Coupled Attention, JMLR 2022.

---

### Official Review · Reviewer_ESqy · 2022-10-23

**Confidence:** 3
**Correctness:** 4
**Technical Novelty And Significance:** 4
**Empirical Novelty And Significance:** 3
**Recommendation:** 8

**Clarity, Quality, Novelty And Reproducibility:**

Paper is clearly written. Lots of details in the appendix, which makes reproducing this paper easier.

**Strength And Weaknesses:**

The kernel manipulation process shown in appendix or "Algorithm" table in page 5 is simple with low computational complexity. Yielding better numerical results.

When the kernel changes (w.r.t. say FNO), the solutions at the internal regions also changes. Authors give a theorem to bound the new solution with respect to the solution without kernel manipulation.

I may have missed it, it seems no bounds for the solutions with respect to the true solution is given.

Paper is evaluated for 1D and 2D systems. Good if evaluation can also be done on 3D systems which is more useful in real world applications.

**Summary Of The Paper:**

This paper uses the neural operator framework with kernel to learn solutions to PDE (see e.g. Neural Fourier Operator work, FNO) as a base method. The paper add modifications to the discretised kernel to achieve better predictions on the boundaries on Dirichlet, Neumann, Periodic boundary conditions.

**Summary Of The Review:**

Generally a good paper, I have not checked through all equations.

Minor points:
1. In Eq.(1), would be more complete if the authors define the mapping of u.
2. Is there a typo when the author define the mapping of kernel right before Eq. (2)?

---

> ### Author Response · Authors · 2022-11-11
> **Response to the reviews and thank you!**
>
> We are thankful to the reviewer for reviewing our work and acknowledging the significance of the results as well as the novelty of the proposed solutions. We answer the specific questions of the reviewer below.
>
> **Bounds** The current paper bounds kernel corrected operator (BOON) with the given base neural operator (e.g. FNO, MWT). Since bounds of NO with true solution are provided in Ref [R2.1], we can simply bound the BOON using triangle inequality. We are happy to include the [R2.1] in our references.
>
> **3D evaluations** We have shown that the current Algorithms easily extend to higher dimensions, which we validate through 1D time-varying, as well as 2D time-varying for all three boundary corrections. The same extension can be easily done for 3D data in future work!
>
> **Minor points** Eq. (1) introduces the boundary value problem and does not concern the operator learning yet. We would like to point out that, in Section 4, when we solve the BVP in various scenarios, the operator map definition is provided in Section 4 para-1 (page 5).
>
> We have verified any possibility of a typo in the paragraph before eq. (2). However, we would be happy to address any specific doubts of the reviewer.
>
> References:
> [R2.1] N. Kovachki et al., “On universal approximation and error bounds for Fourier Neural Operators”, 2021.

---

### Official Review · Reviewer_Z7Rp · 2022-11-02

**Confidence:** 3
**Correctness:** 3
**Technical Novelty And Significance:** 2
**Empirical Novelty And Significance:** 3
**Recommendation:** 5

**Clarity, Quality, Novelty And Reproducibility:**

Clarity:
- Section 2.2: Neural Operators Architectures are barely introduced and the introduction is generally very high-level, leaving out how the integral operators are actually implemented (e.g. in FNO).

- The notation is at times very confusing, and definitions of important parts are scattered around.
E.g. $\mathcal{G}$ is introduced as a function in the BVP setting (1), although for the whole paper $\mathcal{G}$ is assumed to be a linear operator.
Or for understanding the meaning of $\mathcal{U}_{bdy} = \mathcal{U}\cup \delta \mathcal{U}$ one has to search for the definition of $\delta \mathcal{U}$ in Table 1.

- Section 3.2 does not really explain the kernel correction. The explanation for the discretized setting is instead provided in the Appendix, where the correction can be formulated as matrix multiplications. In the main text only the more general algorithm is provided, but without any explanations, leaving the reader confused.

- Figure 2 is a bit confusing. Shouldn't both arrows originate from the same point in $\mathcal{A}$, but with different trajectories as only the operator changes? Also, why is here the operator applied to the error( $T(\mathcal{E}_\text{bdy}) $  )?
Shouldn't it be $\mathcal{E}_\text{bdy}(T)$ so that it is consistent with equation (3)?

Quality:
The experimental settings appear to be sound, as well as the motivation for correcting the kernels.
Experiments however lack repeated runs.
I did not go through the proofs in the appendix in detail.

Novelty:
The paper proposes a general way to hard-code boundary conditions into Neural Operator networks, which has to the best of my knowledge not been done before.


Reproducibility:
- Code is provided
- Experiments appear to be from individual seeds, and lack uncertainty estimates (e.g. standard deviations).

**Strength And Weaknesses:**

Strengths:
1. The proposed kernel correction results in a significant performance improvement in settings where the boundary conditions are known. This is exemplified with a wide range of extensive experiments.

2. The kernel corrections are to my understanding generally applicable to Neural Operator architectures, not just FNOs. In the appendix, an experiment with application to the Multi-Wavelet based Neural Operator is showcased.

3. Code is provided, and it seems to be well documented and maintained.


Weaknesses:
1. There is a lack of discussion on how the setting differs from other work on Neural Operators (e.g. FNO), i.e. what assumptions are made and what limitations the proposed kernel refinement has.
For example: Is this hard-coded approach for boundary conditions still applicable if we consider settings of the same PDE with differing boundary conditions?

2. There are severe clarity issues. E.g. the algorithms for the kernel corrections are provided as pseudo-code, but there is no proper explanation of the underlying idea. One of the reasons is that a "kernel module" $\mathcal{K}$ is suddenly used, which is hand-wavely mentioned in the paper but never properly defined.

General:

I strongly recommend the authors to restructure the paper.
Instead of the general and vague introduction of Neural Operators and the kernel correction, I'd suggest concretely introducing one architecture (e.g. FNO), and how the kernel correction can be applied to it.
Highlighting the generality of the approach can still be done afterwards, but right now the lack of a concrete example is confusing.
Although the extensive experiments are of course valuable, they could partially be moved to the appendix, offering some space for section 3.


**Summary Of The Paper:**

The paper proposes the additional use of boundary conditions for Neural Operator networks.
The boundary conditions are explicitly enforced in the parameterized operator kernel, such that they are in principle fulfilled "by construction".
Dirichlet, Neumann, and Periodic boundary conditions are considered.

For experiments, the proposed method is applied to Fourier Neural Operators (FNO) and compared to state-of-the-art operator learning architectures.
In multiple numerical experiments for PDEs in 1D and 2D (+ Time Dimension) with different boundary conditions,
a significant decrease in relative L2 error is demonstrated compared to vanilla FNO and other architectures.

**Summary Of The Review:**

The paper provides a strong performance improvement to Neural Operator networks in settings where boundary conditions are available.
The novelty lies mainly in additionally considering boundary constraints in existing Neural Operator Architectures.

However, due to the lack of clarity, I vote against accepting this submission.
This is mainly due to the lack of explanation for algorithms 1 to 3 and in general section 3.2.

I'd like to clarify, that the method is a valuable contribution, but would greatly benefit from another iteration.


***
## Edit after Author's Response
The authors resolved with their response a few issues, namely:
- Notation was clarified with additional info provided in the Appendix
- Appendix C.2 was added, which explains the origin of the previously hard-to-understand Algorithms 1-3
- The issue of repeated experiments was resolved, by providing information about very little variation in the experiment results.
- A discussion of the special case for enforcing a boundary value to be zero was added to Appendix B., showing that in principle a solution with any given (but non-zero) precision is still possible.

With that, most of my major concerns have been resolved and I will raise my score.
I still have my doubts regarding the structure of the paper and the usefulness of Algorithms 1-3 in the main part, while all of the actual explanations happen in the Appendix.

---

> ### Author Response · Authors · 2022-11-11
> **Response to the reviews (1/2)**
>
> We thank the reviewer for taking the time to give a detailed read of our paper, and also for recognizing the novelty of our proposed approach, and our experiments as extensive. Following the reviewer’s suggestion, we have improved the clarity of Section-3.2 and also added enhanced explanations for the Algorithms on page 4, as well as Figure 6 (on page 21) for the neural architecture of BOON Algorithms. Our responses to the reviewer’s questions are as follows:
>
> **How our approach differs from other NOs (assumptions/limitations)** (1) To the best of our knowledge, prior Neural Operators do not formalize the boundary value problem (BVP), defined in equation 1, where the boundary conditions are specified as part of the problem definition, and at best rely only on training data to try learning BCs implicitly. Thus, the predicted solutions are not guaranteed to exactly satisfy the BCs as we show in Figure 1 with FNO. In science and engineering problems, boundary conditions are physically meaningful (for example see Figure 1 on the heat insulator boundary condition problem, and our Navier Stokes experiment of lid-cavity driven flow, where the flow itself is determined by the boundary condition). A violation of the BCs can lead to non-physical results as shown in Figure 1.
>
> (2) We wish to clarify that, in our approach, the boundary conditions are not hard-coded. They are provided as input, along with the training data and a Neural Operator. Our method BOON can be integrated into any kernel-based Neural Operator enforcing the given BCs explicitly (as shown in Appendix F.5 with the Multi-wavelet based Operator). For example, hard-coding boundary conditions can only result in improvement over specific boundary points. However, we show in Figures 3 and 4 how our approach guides the operator learning process to not only (a) satisfy the boundary constraints but also (b) learn a better **interior** profile. It is only because of (b) that we are observing an aggregate improvement of 2X-20X in relative L2 error across the entire domain (boundary + interior).
>
> (3) Finally, for the same PDE, changing the BCs yields a different BVP with a **new true solution**, therefore changing the output space $\mathcal{U}$ and hence the operator $T$ as well. A modification in the true solution requires the generation of new training data and therefore a re-training of the models is required. This is true for all existing works like FNO, MWT, and our proposed BOON, since the underlying operator is changed. We refer the reviewer to Sections 4.1.2, and 4.3, where we show the same Burgers’ equation PDE with Dirichlet, and periodic boundary conditions, respectively, and different corresponding initial conditions. We see that the solution and corresponding errors change for all the methods since a *change of boundary conditions is a different problem with a different corresponding solution*.
>
> Through our extensive experiments, we also demonstrate that satisfying boundary conditions with our BOON method do not break other Neural Operator properties. Our model still satisfies the resolution invariance property (see Appendix F.4 on page 28), and it is computationally efficient by using the module $\mathcal{K}$, rather than forming the discrete kernel matrix explicitly (See Theorem 2).
>
> As the reviewer mentioned, this method is applicable when the BCs are known, which is very common in practice as a part of the problem definition. For our assumptions, we currently consider linear types of boundary conditions (Dirichlet, Neumann, periodic), which have been defined in the paper as the linear $\mathcal{G}$ operator. We choose these constraints due to their immense applications in various physics simulations. Extensions to non-linear boundary conditions can be the focus of future work!

---

> > ### Author Response · Authors · 2022-11-11
> > **Response to the reviews (2/2)**
> >
> > **Algorithms/Kernel Module** Thank you for the suggestion to restructure Section 3.2 for improved clarity! We provide additional clarifications in Section 3.2, and a proof of Theorem 2 in Appendix C.2 (page 17), which is a simple algebraic expansion of the already preset proof in Appendix C.1, to help better clarify the algorithms. For the Dirichlet and Neumann boundary conditions, we apply Gaussian elimination to the first row (for left BCs) or last row (for right BCs) of the discrete kernel matrix to enforce the boundary value of the function (Dirichlet) or derivative (Neumann) at a specific boundary.  For periodic boundary conditions, we set the first and last rows of the discrete kernel matrix equal to each other. Figure 5 provides an illustration of the application of these operators. We show that the discrete matrix form only relies on matrix-vector multiplications with the kernel matrix, as standard with Neural Operators. We can then compute these matrix-vector multiplications efficiently by making use of a kernel module $\mathcal{K}(x) \coloneqq Kx$, e.g., Fourier bases in FNO, and multiwavelet bases in MWT to avoid explicitly forming the kernel matrix. This results in a $\mathcal{O}(N_K)$ complexity, where $O(N_K)$ is the complexity of the base neural operator method.
> >
> > **Concrete BOON architecture** Following the reviewers’ suggestion, we have added Figure 6 on page 21 to illustrate in detail the end-to-end BOON architecture, and referenced it in Section 3.2 on page 5. As mentioned in Section 4, page 6 already, we concatenate multiple corrected kernels with non-linearities (e.g., GeLU) in between. The individual ${\bf K}_{\text{bdy}}$ architectures are a straightforward outcome of the proposed Algorithms 1,2, and 3 for Dirichlet, Neumann, and periodic, respectively.
> >
> > Below we address the clarity issues:
> >
> > **Neural Operators Background** Due to space constraints, for Section 2.2, we kindly refer the reviewer to the FNO work (Li et al. (2020a)) for Fourier kernel, and MWT (Gupta et al. (2021)) for computing multiwavelet kernel. We include the main integral operator equation that is needed in our work in the background in eq. (2). Note that an integral computation is not required as we are dealing with the functions evaluated over a finite grid, as mentioned in Section 4 **data generation** first paragraph, page 5. In such cases, the integral reduces to a matrix-vector multiplication.
> >
> > **Notation** We would like to emphasize that $\mathcal{G}$ is an operator and the current work never mentions it as a function. After introducing $\mathcal{G}$ in eq. (1), it has been defined as a linear operator in the paragraph following eq. (1) on page 3. However, the authors believe that the confusion may have arisen from extra parentheses after $\mathcal{G}$ in eq. (1). While this is also an operator notation but for the sake of clarity, we have removed extra parentheses in eq. (1).
> >
> > It should be $\partial\mathcal{U}$, which is defined in Table 1, as the function space corresponding to a given boundary condition in Table 1. We have clarified in the caption of Table 1 and added a summary of the notations used in the paper in Appendix A on page 14. In addition, we updated Figure 2 to provide an illustration of $\partial \mathcal{U}$, $\mathcal{U}$, and $\mathcal{U}_{\text{bdy}}$.
> >
> > **Kernel Correction** In Section 3.2 on page 4, we add an overview of BOON. We also add Appendix C.2 (page 18) to derive the simple algebraic expansions that lead to the exact lines in the algorithm. These derivations clarify that our BOON boundary correction consists of 3 kernel matrix-vector multiplication operators, which can be efficiently computed using a module $\mathcal{K}(x)\coloneqq Kx$ for a given Neural Operator basis. We add a further explanation of this module directly to Section 3.2 as well.
> >
> > **Figure 2** We have corrected Figure 2, thank you! We added an illustration of $\partial \mathcal{U}$ here as well.
> >
> > **Reproducibility** As mentioned in the experimental settings in Section 4, we run all models for 500 epochs with a learning rate decaying by 0.5 after every 100 epochs. This yields a very stable performance (relative L2 error) across multiple seeds, and relative L2 error saturates even after just 300 epochs. Such behavior is consistent with the previous works of FNO and MWT, where they have also made a similar observation. For example, upon running for 3 seeds, for Table 3, we get a standard deviation for BOON as 1.7e–5 (nu=0.1), 4.5e-6 (nu=0.02), 3.1e-7 (nu=0.005), 2.2e-7 (nu=0.002), and 1.9e-6 (nu=0.001). The std. dev. is orders-magnitude lower which suggests stability of results over multiple runs.

---

> > > ### Comment · Reviewer_Z7Rp · 2022-11-17
> > > **Response**
> > >
> > > Thank you for the detailed response and the provided clarifications.
> > >
> > > My concerns regarding reproducibility / repeated runs are resolved.
> > > Some of my misconceptions regarding the encoding of the boundaries were clarified.
> > >
> > > Regarding the issue of clarity, I will have to go through the sections of the adapted paper in detail.

---

> > > > ### Author Response · Authors · 2022-11-18
> > > > **Author comment**
> > > >
> > > > We are glad that the reviewer’s concerns are addressed! We are also hopeful that the updated manuscript, especially Section-3.2, Appendix C.2, and Figure-6, will further clarify the author's remaining concerns about the proposed Algorithms.

---

> > > > > ### Comment · Reviewer_Z7Rp · 2022-11-23
> > > > > **Appendix C.2 clarified a lot**
> > > > >
> > > > > After reading Appendix C.2 I finally follow the reasoning of Algorithms 1-3, and how the actual kernel modification can be applied to arbitrary kernel modules.
> > > > >
> > > > > I still think this should be (in a compact version of course) part of the main paper,
> > > > > as it is somehow required for actually understanding Algorithms 1 to 3.
> > > > >
> > > > > Nonetheless, the paper is now (when including the Appendix) self-contained & understandable, and my main clarity issues have been resolved.
> > > > >
> > > > > I also appreciate the discussion about the corner case of enforcing the boundary values to be zero in Appendix B.
> > > > >
> > > > > Minor Typos I found while rereading:
> > > > > - Proposition 1.  3. Periodic:  At the end of the line is a minus instead of an equal sign.
> > > > > - Appendix B, Periodic (Bottom of Page 15.): The "thanks" in the sentence seems misplaced / does not make any sense.

---

> > > ### Comment · Reviewer_2TdD · 2022-11-24
> > > **response**
> > >
> > > I want to acknowledge the author's efforts in responding to our concerns. My other concerns have largely been adressed, however I still have lots of trouble with clarity.
> > >
> > > This is a machine learning conference, where one should not assume advanced mathematical background for your target audience. Re-reading the paper I am still lost on the notation, which seems to value elegance or being fancy over simplicity. After reading all the responses, and re-reading the paper again, I still have no idea what the U_bdy means or tries to mean. I assume it means "body", which to me sounds like A or U without the border. Yet, fig2 shows something completely else, and I can't decipher what it means. Somehow the border is partially inside U, what...? Similarly, the \partial H norm notation is head-scratching. The sec 2.2. is still given with practically no explanation, and this means that only people already experts in FNO can fully appreciate this work. Furthermore, the main beef of the paper is the constraint satisfaction algorithms 1-3. These can't be understood just from the text. It does not help that one hides the main ideas into the appendix. The notation issues also are not fixed by adding more explanations to appendix.
> > >
> > > I believe the authors should revise the presentation by simplifying the complex notation and bringing helpful illustration into the main paper, and putting less relevant technical stuff to appendix (eg. the algorithms).

---

### Author Response · Authors · 2022-11-11
**Author response**

We are thankful to all the reviewers for giving a careful read to our paper and providing thoughtful comments and suggestions, which have led us to further improve the presentation of our work. It is very encouraging to see that all reviewers have acknowledged the novelty of our approach and the significance of our results! For example, Reviewer Z7Rp states that our kernel correction method has “not been done before”.  Reviewer-2TdD lauds the “good novelty” of our “principled method”, and Reviewer-ESqy calls our work a “solid paper that successfully includes proper boundary behavior to FNO models”.  We provide an individual response that clarifies each reviewer’s concerns, and a summary of the changes made to the paper is listed below.

1) **Algorithms clarification** We have added Appendix C.2 on page 18 to further elaborate the algebraic operations that lead to Algorithms 1,2,3 from Theorem 1.

2) **Enhancing Section 3.2** We have added some more details to better connect the existing theoretical details in the Appendices to the main text of Section 3.2.

3) **Kernel correcting architecture** Figure 6 is added on page 21 which directly translates the existing Algorithm 1,2,3 into a neural architecture for a detailed understanding of the flow, and is referenced in Section 3.2.

4) **Notations** We have added a comprehensive notations table in Appendix A on page 14 to unify the notations in the complete paper.

---

### Decision · Program_Chairs · 2023-01-20

**Decision:**

Accept: poster

**Justification For Why Not Higher Score:**

In the end, I think that this is a solid application-oriented paper, but I am also convinced that this paper will not revolutionize the field.

**Justification For Why Not Lower Score:**

Well motivated method and convincing experimental evaluation -> a solid paper

**Metareview: Summary, Strengths And Weaknesses:**

For this paper, we have two "borderline" reviews, and one clearly positive review. The more neutral reviewers mentioned several potential weaknesses, such as the unclear novelty, lacking explanation of algorithmic ideas and relevance to the machine learning community. Some of these concerns could be addressed in the rebuttal, but the two reviewers finally wanted to stick to their borderline scores.
On the positive side,  most reviewers agreed that the method proposed works well in many experiments, sometimes indeed leading to significant performance improvements (if the boundary conditions are known), that the provided code is clear and well documented, and that it is easy to reproduce the results.
In summary, this paper might still be discussed in a controversial way, but putting all the reviews, the rebuttal and the discussions together, for me the positive aspects dominate. Therefore I vote for acceptance.

**Note From Pc:**

if the above contains the word "oral" or "spotlight" please see: "oral" presentation means -> notable-top-5% and "spotlight" means -> notable-top-25%. As stated in our emails, we are disassociating presentation type from AC recommendations